# Monometallic interphasic synergy via nano-hetero-interfacing for hydrogen evolution in alkaline electrolytes

Kamran Dastafkan [1], Xiangjian Shen [2], Rosalie K. Hocking [3], Quentin Meyer[1] & Chuan Zhao [1] ✉

Electrocatalytic synergy is a functional yet underrated concept in electrocatalysis. Often, it materializes as intermetallic interaction between different metals. We demonstrate interphasic synergy in monometallic structures is as much effective. An interphasic synergy between $Ni(OH)_2$ and Ni-N/Ni-C phases is reported for alkaline hydrogen evolution reaction that lowers the energy barriers for hydrogen adsorption-desorption and facilitates that of hydroxyl intermediates. This makes ready-to-serve Ni active sites and allocates a large amount of Ni $d$-states at Fermi level to promote charge redistribution from $Ni(OH)_2$ to Ni-N/Ni-C and the co-adsorption of $H_{ads}$ and $OH_{ads}$ intermediates on Ni-N/Ni-C moieties. As a result, a $Ni(OH)_2$@Ni-N/Ni-C hetero-hierarchical nanostructure is developed, lowering the overpotentials to deliver −10 and −100 mA cm$^{-2}$ in alkaline media by 102 and 113 mV, respectively, compared to monophasic $Ni(OH)_2$ catalyst. This study unveils the interphasic synergy as an effective strategy to design monometallic electrocatalysts for water splitting and other energy applications.

The development of efficient noble metal-free electrocatalysts for electrochemical water splitting requires the understanding of different activity enhancement factors. Ensemble effects brought by synergistic interactions between two or more structural entities on Faradaic electron transfer process can be induced by cumulative surface active sites, interfacial charge transfer, and modulating electronic structure. Understanding the origin and mechanism of the impact of these synergistic interactions on the oxygen evolution reaction (OER) and hydrogen evolution reaction (HER) has remained elusive. So far, electrocatalytic synergy has been mainly studied for metallic dopants[1–5], host-guest atomic interactions in multi-metallic structures[6–9], heterostructures[10–12], and multi-component compounds[13–15]. In most cases, synergistic effects are highly dependent on composition and electronic structure.

Intermetallic synergy is mostly realized, when a foreign metallic dopant is introduced[16,17]. The most well-known example is Ni-Fe synergy for OER which has marked NiFe-based (oxy)hydroxide as a benchmark OER catalyst in alkaline media[8]. The enhancement in intrinsic activity, accelerated interfacial charge transfer and in situ charge transformation, increased electroconductivity, as well as modulated electronic structure are reported not only via promoted bimetallic Ni-Fe and Co-Fe synergy, but also through establishing auxiliary synergy with other added metals[18–21].

In comparison, electrocatalytic synergy is less pronounced for HER. A well-known example is between Ni and Mo, with the corresponding alloy structures giving the highest electrocatalytic efficiency for HER, comparable to Pt/C catalysts[22,23]. Whether how much the bonding structure, electron transfer pathway, or the

[1]School of Chemistry, UNSW Materials and Manufacturing Futures Institute, The University of New South Wales, Sydney, New South Wales 2052, Australia. [2]Engineering Research Center of Advanced Functional Material Manufacturing of Ministry of Education, Zhengzhou University, Zhengzhou 450001, China. [3]Department of Chemistry and Biotechnology, Centre for Translational Atomaterials and ARC Training Centre for Surface Engineering for Advanced Material SEAM, Swinburne University of Technology, Hawthorn, VIC 3122, Australia. ✉e-mail: chuan.zhao@unsw.edu.au

amount of adsorption sites on the catalyst surface are individually affected, the synergistic interaction (*i.e.*, between Ni and Mo) augments intrinsic activity by expediting the HER kinetics[24–26]. This involves accelerating water dissociation and hydrogen (and/or hydroxyl species in alkaline pH) adsorption-desorption to achieve faster hydrogen recombination[11,24]. In this context, the synergistic interactions in monometallic systems are not completely understood. As a major active site for HER, promoting the intrinsic activity of Ni by synergy is a fundamentally fascinating and practically important approach.

Herein, we report an interphasic synergy in a monometallic structure between Ni(OH)$_2$ and Ni-N/Ni-C phases which boosts the HER activity significantly at both low and high current densities. Metallic Ni and Ni(OH)$_2$ are recognized as excellent water dissociation promoters but have too strong hydrogen affinity which is not beneficial for hydrogen recombination[27–30]. Ni non-oxides including Ni-N-C structures on the other hand do not have optimal energy for H-OH cleavage despite having labile electronic properties such as charge polarization[31–34]. We show a hetero-hierarchical nanostructure grown on a Ni foam (Ni(OH)$_2$@Ni-N/Ni-C/NF) in which surface-localized interfacing accelerates H$_2$ recombination (Heyrovsky or Tafel) following the promoted water dissociation on Ni(OH)$_2$. Systematic electrochemical and spectroscopic measurements as well as density functional theory calculations reveal that the interphasic synergy procures ready-to-serve Ni active sites and maintains the promoted intrinsic activity under a wide range of HER potentials.

## Results

The Ni(OH)$_2$@Ni-N/Ni-C/NF electrode is prepared via a two-step synthesis involving solvothermal reaction using Ni foam as starting material and thermal treatment, where the solvothermal reaction is optimized by response surface methodology (RSM) (See methods, Figs. S1–3, Table S1–3, Supplementary Note 1). Ethylenediamine (EDA) was found to be the most effective among a series of N-containing organic structure directing regents (Figs. S4 and S5). The self-crystallization and decomposition of [Ni(en)$_3$]$^{2+}$ provides Ni-N/Ni-C moieties on the surface of NF. Considering the mutual interactions between the input parameters, the optimal conditions give rise to a hierarchical morphology with localized nano-hetero-interfaces between oxide and non-oxide Ni at nanoscale in a random fashion (Fig. 1a). Anisotropic architectural zones with different growth orientations are detected at the surface of

Ni(OH)$_2$@Ni-N/Ni-C/NF (Fig. 1b, Fig. S6 and S7), while Ni(OH)$_2$/NF has a typical micro/nanosheet morphology (Fig.S8).

The nano-heterostructure forms an ultrathin film over surface hierarchies, captured by focused ion beam-scanning transmission electron microscopy (Figs. S9 and S10). A weak polycrystallinity is detected with (110) and (116) crystal planes, due to graphitic carbon, as well as (111) and (112) planes ascribed to NiC$_{10}$ and Ni(OH)$_2$, respectively (Fig. 1c). Randomly formed Ni(OH)$_2$ and Ni-C moieties in an outer carbon layer indicate a weak short-range order (Fig. 1d). Limited NiC$_{10}$ crystalline phases within the carbon layer depict interplanar spacing for (204) and (114) planes, corresponding to the randomly selected (R$_2$) and (R$_3$) regions, respectively. Ni(OH)$_2$ phase is detected in the vicinity with the assigned (101) crystal planes. The represented fast Fourier transforms of (R$_2$) and (R$_3$) regions indicate the formation of local nano-hetero-interfaces, while they are disrupted in bulk areas with more uniformity, *i.e.*, (123) crystal plane of NiC$_{10}$ (R$_1$) (Fig. 1d inset). Elemental mapping reveals the increasing distribution of O, C, and N atoms near the surface, suggesting the emergence of localized N-C and Ni-N bonds (Fig. 1e and Fig. S11). The poor crystallinity and short-range order are enhanced toward the surface, where a carbon shell with (101) plane (R$_4$) is detected. Thus, nano-hetero-interfaces are inlaid in an ultrathin layer over the crests and valleys of surface hierarchies instead of a compact uniform heterostructure. The formation of the amorphous carbon layer promotes nano-hetero-interfacing by encasing and stabilizing Ni structures in the vicinity and at short-range order as well as by boosting the electronic conductivity. The random localization at surface and short-range order as well as the carbon shell endow an overall amorphicity (Fig. S12).

The locally emerged nano-hetero-interfaces at the surface of NF promote HER significantly in alkaline electrolytes. Figure 2a and Fig. S13 show that low overpotentials of 60, 141, and 223 mV are derived by the optimal nano-heterostructure to deliver current densities of −10, −100, and −1000 mA cm$^{-2}$, respectively, which are 69, 113, and 177 mV lower than those obtained with monophasic Ni(OH)$_2$/NF. The same trend is observed when no external Ni ions are supplied (Fig. S14). Intriguingly, comparing the electrochemical activation to get HER polarization elucidates both faster stabilization of HER activity and significant shifts between LSV curves at onset and high potential regions for Ni(OH)$_2$@Ni-N/Ni-C/NF, whereas it takes more activation for Ni(OH)$_2$/NF to attain stable LSV curves and with no significant change (Fig. S15). Tafel slope is decreased to 43.9 mV dec$^{-1}$ at low

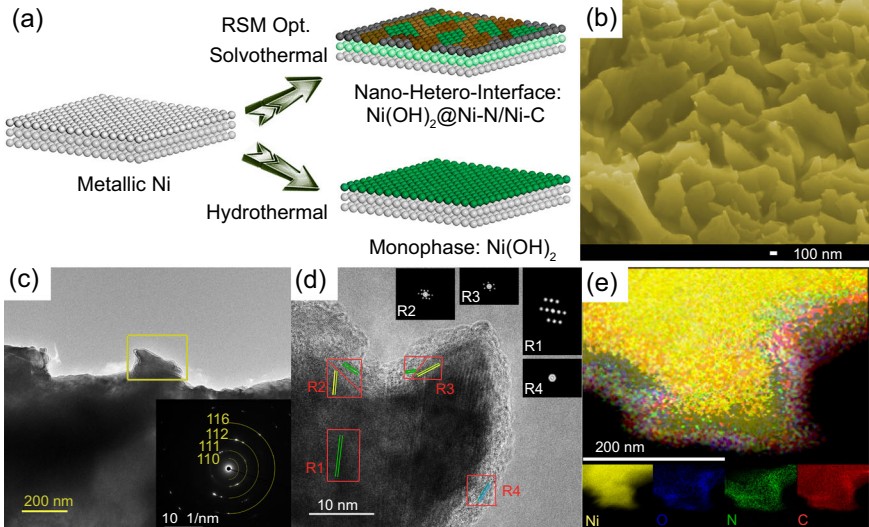

**Fig. 1 | Morphology and structure by nano-hetero-interfacing. a** Fabrication of Ni(OH)$_2$@Ni-N/Ni-C nano-hetero-interfaces on NF using RSM optimization. **b** High-resolution SEM, (**c**) and (**d**) low- and high-resolution TEM, (**c** inset) selected area electron diffraction pattern, (**d** insets) fast Fourier transforms of the selected regions, (**e**) elemental mapping of surface hierarchies of Ni(OH)$_2$@Ni-N/Ni-C/NF.

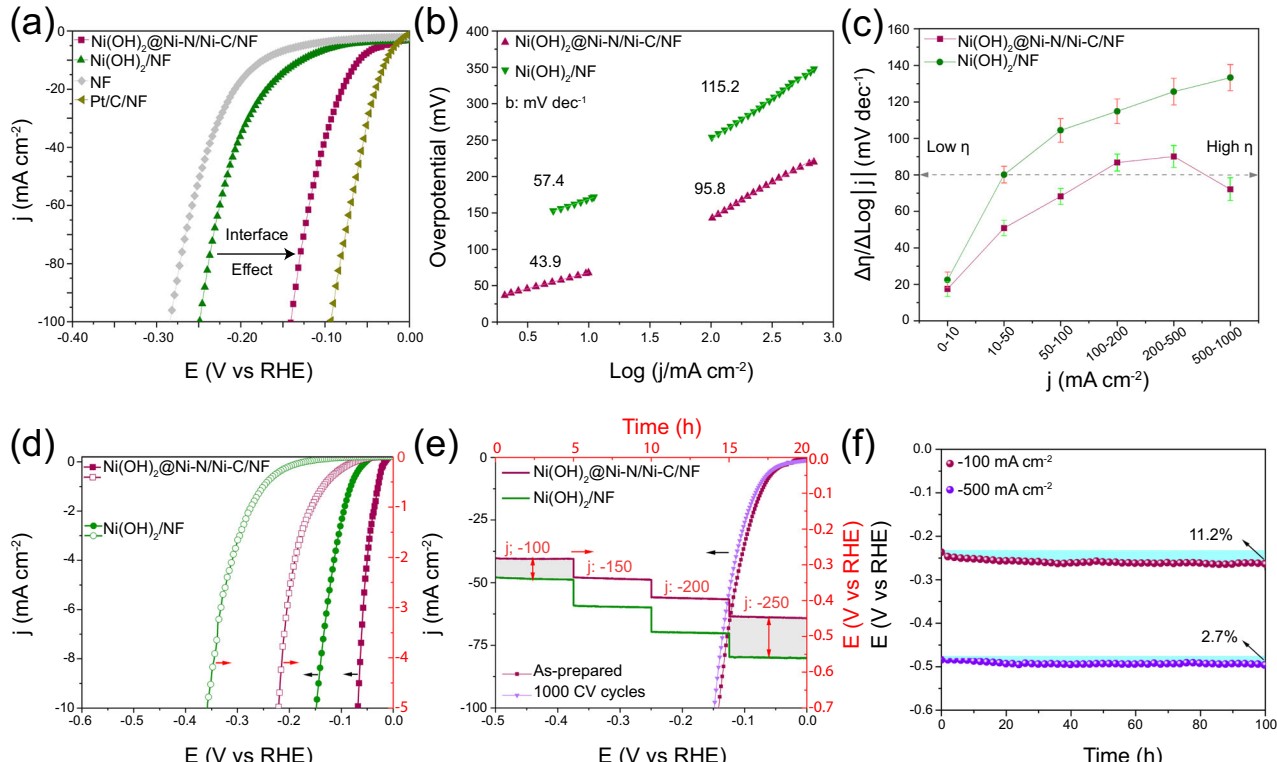

**Fig. 2 | HER performance. a** HER polarization curves and (**b**) Tafel slopes in 1 M KOH. (**c**) Plot of ratio of Δη/ΔLog|j| (η: overpotential) as a function of current density. Error bars represent standard deviation. **d** ECSA-corrected polarization curves (hollow) against apparent activity (solid). **e** Multistep chronopotentiometry of Ni(OH)₂@Ni-N/Ni-C/NF and Ni(HO)₂/NF as well as HER polarization of Ni(OH)₂@Ni-N/Ni-C/NF before and after 1000 CV cycles. **f** Stability of Ni(OH)₂@Ni-N/Ni-C/NF at fixed current densities.

current densities up to −10 mA cm⁻¹, compared to 57.4 mV dec⁻¹ for Ni(OH)₂/NF, revealing the boosted electrochemical desorption of hydrogen at Volmer-Heyrovsky pathway. The accelerated reaction rate is even more distinct at large current densities above 100 mA cm⁻² with a slope of 95.8 mV dec⁻¹, compared to 115.2 mV dec⁻¹ for Ni(OH)₂/NF (Fig. 2b). These results put Ni(OH)₂@Ni-N/Ni-C among the best bimetallic and multiphasic Ni-based HER catalysts in alkaline media to date (Fig. S16 and Table S4).

The Δη/ΔLog|j| ratio is further used as an indicator to evaluate the combined effects of electron transfer and mass transport on HER performance[35]. Figure 2c shows a much smaller increase of Δη/ΔLog|j| ratio for Ni(OH)₂@Ni-N/Ni-C/NF at a wide range of current density and even a decrease at large current densities. The more prominent decrease in the overpotential difference between the large current densities of 500 and 1000 mA cm⁻² compared to the impact of the current density difference reduces the Δη/ΔLog|j| ratio, implicating the retained intrinsic activity and accelerated HER kinetics at large current densities for Ni(OH)₂@Ni-N/Ni-C/NF. In comparison, the monophasic Ni(OH)₂/NF experiences an increasing trend with current density. The electrochemically active surface area of Ni(OH)₂@Ni-N/Ni-C/NF is slightly smaller than that of Ni(OH)₂ (Fig. S17). However, intercoupling of many locally formed short-ranged phases render more utilizable surface-active sites and hence higher intrinsic activity (Fig. 2d). The trend of electrocatalytic turnover frequency (TOF) with HER overpotential depicts the significantly boosted intrinsic activity of Ni(OH)₂@Ni-N/Ni-C/NF (Fig. S18). Specifically, at an overpotential of 100 mV, Ni(OH)₂@Ni-N/Ni-C/NF has a TOF of 0.402 H₂ s⁻¹, which is 6.8 times higher than that of Ni(OH)₂/NF (0.059 H₂ s⁻¹). Ni(OH)₂@Ni-N/Ni-C/NF depicts durable and stable performances with incrementing current difference with monophasic Ni(OH)₂ during multistep chronopotentiometry, small potential difference after 1000 CV cycles, as well as long term chronopotentiometry at large current densities

(Fig. 2e, f). The surface hierarchies are preserved after 100 h of HER at −100 and −500 mA cm⁻² (Fig. S19 and S20). Nano-hetero-interfacing is also preserved with partial disruption early on in HER (Fig. S21a) but debilitated with extended amorphization after long-term HER performance at −100 mA cm⁻² (Fig. S21b). The generated H₂ gas was quantified using gas chromatography which gave Faradaic efficiencies close to unity at different current densities (Fig. S22).

Modulating monometallic electronic structures and the underlying interphasic synergy is a universal strategy for developing new nonprecious metal-based catalysts. In fact, the nano-hetero-interfacing of Ni(OH)₂, as a good water dissociation promoter but with overly strong affinity for hydrogen, with Ni-N/Ni-C phase entails even more pronounced differences in HER electrocatalysis with monophasic Ni(OH)₂ in acidic media. Hydrogen intermediate following the Volmer step forms through the discharge of protons in acidic environment. Therefore, accelerating the hydrogen adsorption-desorption-recombination process is the primary activity descriptor at low pH. A similar trend in activity enhancement is observed by Ni(OH)₂@Ni-N/Ni-C/NF in 0.5 M H₂SO₄ solution, although with a lower intrinsic activity of Ni in acidic media (Fig. S23a). Compared with the meagre boost in activity by monophasic Ni(OH)₂ with respect to bare Ni foam, nano-hetero-interfacing decreases the Tafel slope and overpotentials to deliver current densities of −10 and −100 mA cm⁻² significantly (Fig. S23b, c). Compared with the overpotential difference obtained in alkaline medium, the larger difference in acid signifies the important role of interphasic synergy in boosting the intrinsic activity of Ni (Fig. S24).

To understand the origin of the enhanced HER performance, the electronic structure is first studied by X-ray absorption near edge structure (XANES) spectra. To avoid self-absorption by metallic Ni and photo-damage, carbon fiber paper (CFP) was used as the catalyst support during the test (See methods). A pre-edge corresponding to

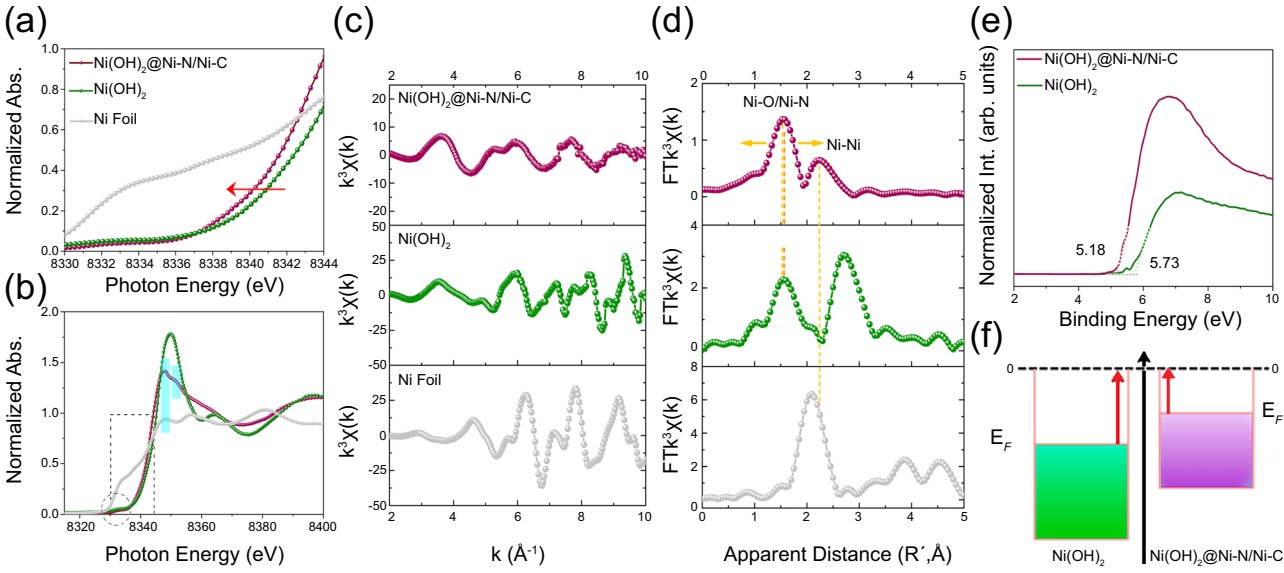

**Fig. 3 | Electronic structure. (a, b)** XANES spectra, **(c)** Ni K-edge EXAFS, and **(d)** Fourier transforms of EXAFS of Ni(OH)$_2$@Ni-N/Ni-C, Ni(OH)$_2$, and Ni foil. **(e)** Valence band spectra and **(f)** schematic work functions of Ni(OH)$_2$@Ni-N/Ni-C/NF and Ni(OH)$_2$/NF.

dipole-forbidden, quadrupole-allowed transition ($1s \rightarrow 3d$) centers occurs at 8333 eV in the Ni K-edge of Ni(OH)$_2$@Ni-N/Ni-C and Ni(OH)$_2$ (Fig. S25)[36]. A shift to lower energy is observed at the absorption edge and white line of Ni(OH)$_2$@Ni-N/Ni-C, implying the altered electronic fine structure of Ni (Fig. 3a, b). The extended X-ray absorption fine structure (EXAFS) displays damped, distorted, and shifted oscillation of $k^3\chi(k)$ function compared to Ni(OH)$_2$ (Fig. 3c). The Fourier transform in radial space (FT-EXAFS) depicts damped and broader peak components between 1 and 2 Å due to a conjunct Ni-O/Ni-N scattering pair, consistent with an altered chemical state of Ni at the local interface between the two phases (Fig. 3d). The Ni-Ni scattering pair lies between those of monophasic Ni(OH)$_2$ and Ni foil, suggesting that inter-coupling with Ni-N/Ni-C moieties endows metallicity to the nano-heterostructure. The EXAFS data fits well with the scattering paths in Ni(OH)$_2$@Ni-N/Ni-C (Fig. S26 and Table S5).

Linear combination analysis (LCA) is undertaken to assimilate the nano-hetero-interfacing. Comparing the XANES spectrum of Ni(OH)$_2$@Ni-N/Ni-C with the linear combination of monophasic Ni(OH)$_2$ and metallic Ni components depicts distinctions over pre-edge, edge, white line, and EXAFS region (Fig. S27a). Whereas the XANES spectrum of Ni(OH)$_2$ matches well with the linear combination of standard Ni(OH)$_2$ and metallic Ni (Fig. S27b). This suggests the distinctive Ni(OH)$_2$@Ni-N/Ni-C structure is emerged from nano-hetero-interfacing of Ni(OH)$_2$ and Ni-N/Ni-C phases, rather their physical combination.

While the hydrothermally grown monophasic Ni(OH)$_2$ micro/nanosheets match well with standard Ni(OH)$_2$ (Fig. S28), inter-coupling of Ni(OH)$_2$ and Ni-N/Ni-C phases holds key featured differences at Ni edge and white line positions along with EXAFS structure with Ni phthalocyanine (Ni-pc) as a control structure for Ni-N/Ni-C moieties (Fig. S29). The index pre-edge feature at 8333 eV is of the same intensity, implying similar $3d–4p$ orbital hybridization of central Ni atoms and the dominance of octahedral symmetry[37]. Meanwhile, there is no splitting of p orbitals like that observed in the characteristic dipole-allowed $1s \rightarrow 4p_z$ transition in the D$_{4h}$ symmetry of Ni-pc (peak I), which remarks a square-planar M−N$_4$ entity (Fig. S29a, b)[36]. In addition to the notable higher energy of Ni edge and white line, Ni(OH)$_2$@Ni-N/Ni-C shows an opposite relative intensity ratio of peaks II and III, which are attributed to $1s \rightarrow 4p_{x,y}$ transition and consecutive scattering, respectively[38]. The enhanced II to III intensity has been correlated with improvement in

electrocatalytic activity[36,39]. Having a different EXAFS pattern (Fig. S29c), FT-EXAFS spectra reveal a shift to shorter apparent distance (1.59 Å) for Ni(OH)$_2$@Ni-N/Ni-C due to the similar scattering paths of Ni-N and Ni-O pairs, as well as appearing of Ni-Ni scattering which is absent in Ni-pc (Fig. S29d).

High-resolution Ni $2p_{3/2}$ X-ray photoelectron spectroscopy (XPS) spectrum deconvolutes into peaks at 854.6 and 855.7 eV along with two shakeup satellites at 860.7 and 864.6 eV (Fig. S30a). The first binding energy shows 0.5 eV and 0.3 eV negative shift compared to those of Ni(OH)$_2$/NF (Fig. S31a) and reported for Ni(OH)$_2$[27]. The second feature shows 0.7 eV negative shift compared to the reported values for Ni-N-C structures[33,40]. This suggests an electron-rich characteristic via intercoupling of Ni phases. High-resolution C 1$s$, N 1$s$, and O 1$s$ XPS profiles further verify the formation of Ni(OH)$_2$ and Ni-N/Ni-C phases (Fig. S30b−d). Particularly, the Ni-O bonding with high-coordination oxygen is more pronounced in Ni(OH)$_2$@Ni-N/Ni-C/NF compared to Ni(OH)$_2$/NF (Fig. S31b). Similar to the shift of Ni K-edge in the XANES of Ni(OH)$_2$@Ni-N/Ni-C to lower energy compared to the monophasic Ni(OH)$_2$ structure, the Ni 2$p$ features in the corresponding XPS spectrum also shifts to lower binding energy (Fig. S32). The observed shifts to lower energy allude to the enhanced electron density and endowed metallicity because of the altered Ni coordination environment and boosted interphasic charge transport at the local interface between Ni(OH)$_2$ and Ni-N/Ni-C phases.

Raman spectroscopy elucidates one- and two-phonon Ni-O, Ni-N, signature D, G, and $C_{2D}$ graphitic carbon vibrations compared with the Raman spectrum of Ni-pc (Fig. S33 and S34a). The D-band around 1335 cm$^{-1}$ and G-band around 1566 cm$^{-1}$ represent the defective and crystalline graphitic structure, respectively. The intensity ratio of D and G bands, depicted as $I_D/I_G$, is therefore an indicator of the degree of graphitization, in that low-intensity ratios ($I_D/I_G < 1$) reveal a high degree of graphitization while high-intensity ratios ($I_D/I_G > 1$) refer to a defective and disordered graphitic structure inclining to amorphicity[41]. Thus, the slightly higher intensity of D band than G band in the optimal nano-heterostructure with added ionic Ni in the solvothermal reaction with an $I_D/I_G$ ratio of 1.12 indicates a defective and amorphous graphitic network, while Ni(OH)$_2$/NF only depicts broad and weak Ni-O vibrations. The effect of Ni ion concentration and RSM optimization is indicated by the weakening of two-phonon Ni-O and graphitic carbon vibrations (Fig. S34b) and the inferior HER performance of the non-optimal heterostructure (Fig. S35). A negative shift by 0.55 eV in the

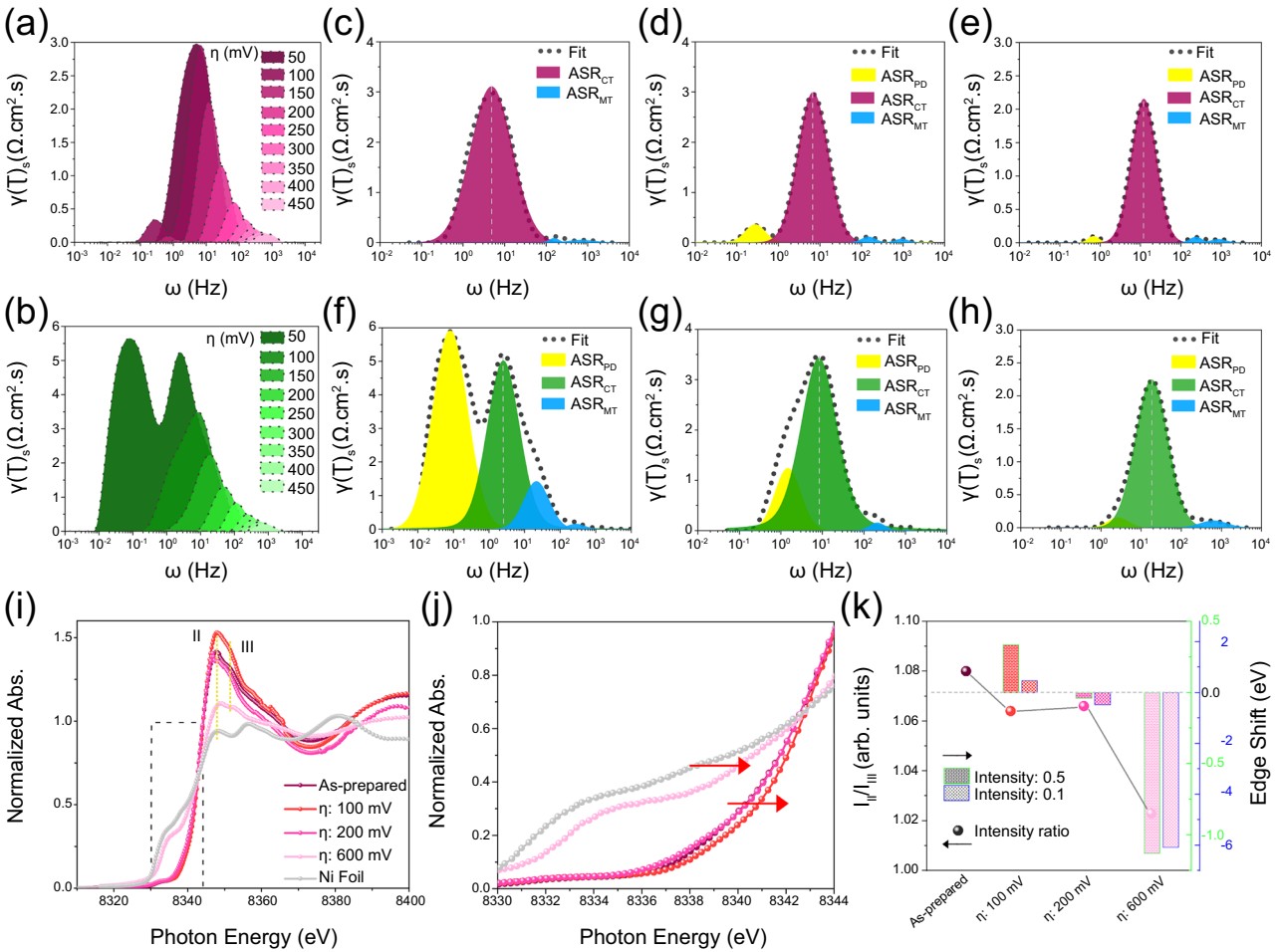

**Fig. 4 | In situ monitoring of charge transfer and structure.** (**a**) and (**b**) DRT plots of Ni(OH)$_2$@Ni-N/Ni-C/NF (purple) and Ni(OH)$_2$/NF (green). DRT deconvolution at overpotentials of (**c**) and (**f**) 50 mV, (**d**) and (**g**) 100 mV, (**e**) and (**h**) 150 mV. Operando (**i**) XANES spectra and (**j**) edge shifts of Ni(OH)$_2$@Ni-N/Ni-C. (**k**) Variation of intensity ratio of peaks II and III as well as edge shifts at intensities of 0.5 and 0.1 in the XANES spectrum of Ni(OH)$_2$@Ni-N/Ni-C.

valence band maximum (VBM) remarks a closer valence band and a closer $d$-band center ($\varepsilon_d$) to Fermi level for Ni(OH)$_2$@Ni-N/Ni-C/NF (Fig. 3e)[42,43]. The upshift of $\varepsilon_d$ for Ni sites loosens the electron binding restriction and reduces the work function (Fig. 3f)[43]. This suggests that nano-hetero-interfacing enhances the energy level of Ni 3$d$ orbitals and facilitates charge redistribution around Ni centers, as well as electron conductivity.

The significant impact of nano-hetero-interfacing on interfacial charge transfer between catalyst and electrolyte is further investigated by in situ electrochemical impedance spectroscopy (EIS, Fig. S36). Plateau-like effective charge transfer ($R_{eff, CT}$) and mass transport ($R_{eff, MT}$) are obtained for Ni(OH)$_2$@Ni-N/Ni-C/NF compared to the sharp decline for Ni(OH)$_2$/NF (Fig. S37). The obtained trends elucidate how effective interface is formed for Faradaic and capacitive processes. Phase relaxation at high, middle, and low frequencies in Bode-phase plots is attributed to charge transport to the interface of catalyst-electrical double layer, intermediate *H adsorption, and Faradaic electron transfer, respectively[6]. The phase relaxation occurs smoothly at mid frequencies over Ni(OH)$_2$@Ni-N/Ni-C/NF with the increase of overpotential (Fig. S38a). In comparison, the phase angles are similar at 50 mV and 100 mV over Ni(OH)$_2$/NF, and the phase relaxation is shifted toward higher frequency, alluding to the poor interfacial interaction of reaction intermediates with Ni sites (Fig. S38b). Also, a significant low-frequency phase shift is observed for Ni(OH)$_2$/NF, pertaining to the sluggish charge transfer due to hydrogen recombination at low overpotentials.

The dynamics of interfacial processes are further investigated by distribution of relaxation times (DRT) analysis of the Nyquist plots from in situ EIS (See methods). The DRT analysis allows the function of time features of interfacial processes to be elucidated, and thus convert the real and imaginary impedance to discrete resistance distribution against frequency[44–47]. Smaller and less altering area specific resistance (ASR) is illustrated for Ni(OH)$_2$@Ni-N/Ni-C/NF from low to high overpotentials, hinting at a ready-to-serve surface (Fig. 4a, b). Deconvolution of DRT peaks decouples interfacial processes to distinct time constants associated with surface mass transport (ASR$_{MT}$), charge transfer due to intermediate adsorption (ASR$_{CT}$), and post diffusion related to H$_2$ recombination (ASR$_{PD}$). ASR$_{CT}$ is the dominant process and rate-determining step (RDS) over Ni(OH)$_2$@Ni-N/Ni-C/NF (Fig. 4c–e). The absence of ASR$_{PD}$ at the beginning of HER points out to the intact hetero-hierarchical nanostructure. The appearance of this peak with small intensity at 100 mV aligns with the partial disruption of nano-hetero-interfaces. In contrast, ASR$_{PD}$ dominates over Ni(OH)$_2$/NF early on in HER, suggesting H$_2$ recombination becomes the RDS in the kinetically controlled region (Fig. 4f). ASR$_{CT}$ re-dominates with increasing the overpotential while ASR$_{PD}$ remains significant (Fig. 4g, h), also referring to the less effective interfacial charge transfer over Ni(OH)$_2$/NF at large overpotentials. The potential-dependent ASR variation indicates a lower kinetic barrier for Ni(OH)$_2$@Ni-N/Ni-C/NF and denotes the impact of an interphasic synergy between Ni phases as the activity enhancement factor

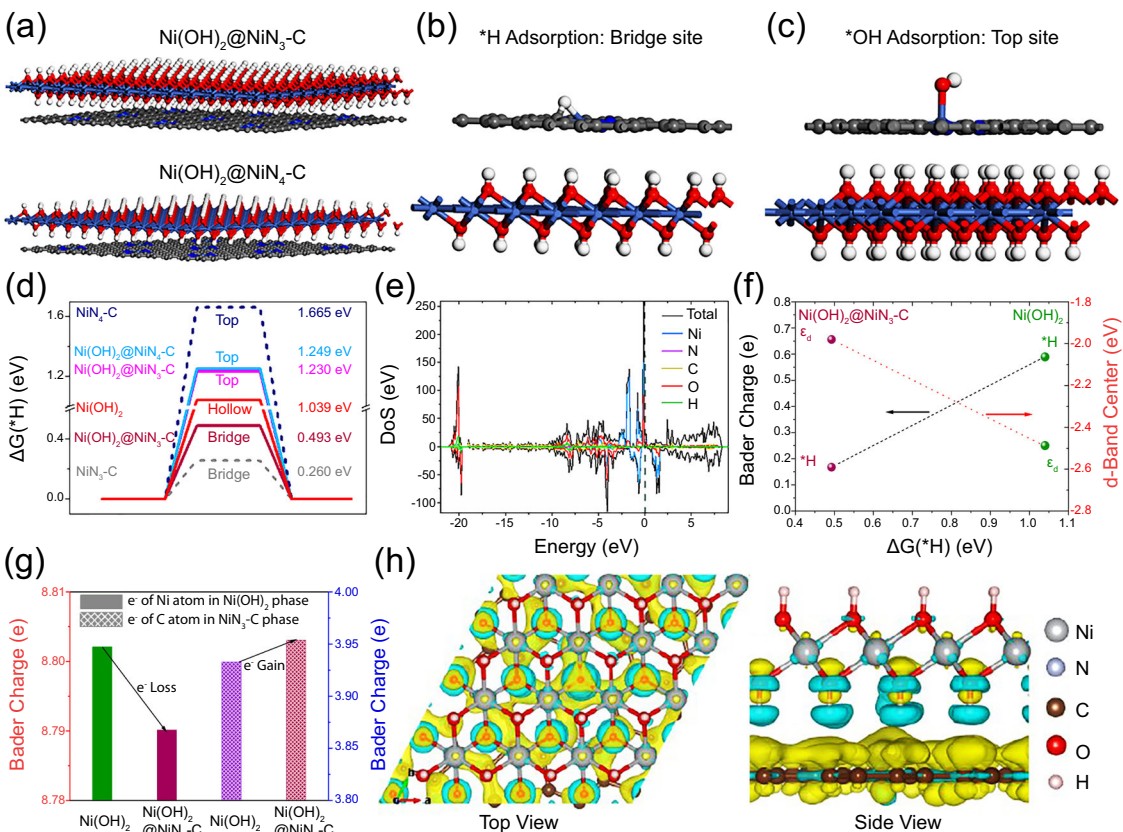

**Fig. 5 | Theoretical simulation of nano-hetero-interfacing. (a)** Atomic structure models of Ni(OH)$_2$@NiN$_x$-C with two Ni-N coordination. Preferred **(b)** *H and **(c)** *OH intermediate adsorption on Ni(OH)$_2$@NiN$_3$-C. **(d)** Free energy diagram of hydrogen adsorption. **(e)** PDOS profile of Ni(OH)$_2$@NiN$_3$-C. **(f)** Relationship between Bader charge of *H adsorption and *d*-band center with the free energy of Ni(OH)$_2$@NiN$_3$-C and Ni(OH)$_2$. **(g)** Illustration of interphasic charge transport via Bader charge variation between Ni(OH)$_2$ and NiN$_3$-C phases and Ni(OH)$_2$@NiN$_3$-C. **(h)** CDD profiles of Ni(OH)$_2$@NiN$_3$-C.

(Fig. S39a). In contrast, monophasic Ni(OH)$_2$ has a low intrinsic activity, and the activity enhancement is only attained by increasing the energy input (Fig. S39b).

The dynamic evolution of the nano-hetero-interfaces during HER is further tracked by quasi-operando XAS (See methods). A similar pre-edge at 100 mV indicates the preserved site symmetry for Ni centers pertaining to $1s \rightarrow 3d$ transition (Fig. S40). Whereas a shift to higher energy at the edge and an intensified white line in the XANES along with slightly disarranged EXAFS k$^3$χ(k) oscillation allude to the partial disruption of surface nano-hetero-interfaces (Fig. 4i, j, Fig. S41). Fitting the EXAFS holds a good correlation with the scattering paths of interfaced Ni phases (Fig. S42a, b, Table S6). The corresponding FT-EXAFS depicts further peak broadening and slight shifts to lower radial distance for Ni-O/Ni-N and Ni-Ni scattering pairs (Fig. S42c). Post-HER core-level Ni 2$p$ XPS spectra at 100 mV at different time periods reveal attenuated Ni 2$p_{1/2}$ and 2$p_{3/2}$ peaks and shakeup satellites as well as increased metallic Ni due to partial catalyst loss at surface and the exposure of the bulk metallic Ni. Interestingly, a negligible shift is observed for the deconvoluted Ni 2$p_{3/2}$ at 854.6 eV while a considerable shift to higher binding energy is observed for the deconvoluted Ni 2$p_{3/2}$ at 855.7 eV, alluding to the partial disruption of the Ni phases along with the size decrease of Ni-N/Ni-C moieties (Fig. S43). That is, Ni atoms are partially oxidized in Ni-N/Ni-C phase due to surface energy and structural defects[48]. In comparison, Ni(OH)$_2$ partially reduces at 100 mV having a Ni K-edge shifted to lower energy (Fig. S44a, b). Disarranged EXAFS k$^3$χ(k) oscillation instigates shift to higher radial distance of Ni-O scattering pair (Fig. S44c, d). The corresponding EXAFS fitting and LCA depict consistent structure with Ni(OH)$_2$ (Fig. S45 and S46).

With increasing the overpotential to 200 mV, Ni sites are slightly reduced and reveal similar state to Ni(OH)$_2$@Ni-N/Ni-C, although similarly disarranged and suppressed EXAFS oscillations still suggests partially disrupted phases (Fig. S41). Ni sites are further reduced at 600 mV. However, the higher XANES edge energy and white line intensity compared to Ni foil, along with more conformity with EXAFS at lower overpotentials suggests the existence of the separated phases with the predominance of Ni(OH)$_2$ even at high overpotentials. Oper-ando XAS also reveals a slight decrease in the peak II to III intensity ratio (Fig. 4i) up to 200 mV, followed by a sharp decline at 600 mV (Fig. 4k). This indicates that the change in interphasic synergy is consistent with the changes in Ni chemical state, as illustrated by Ni K-edge shift (Fig. 4k). Thus, intrinsic activity decline is trivial at least at kinetically-controlled HER region up to 200 mV. Furthermore, conducting LCA at these conditions elucidates similar XANES and linear combinations with the as-prepared Ni(OH)$_2$@Ni-N/Ni-C, compared to metallic Ni at 100 and 200 mV. This indicates negligible attenuation in nano-hetero-interfacing and hence in interphasic synergy (Fig. S47a, b). Even the XANES at 600 mV shows discernible differences with the linear combination of Ni(OH)$_2$@Ni-N/Ni-C and Ni foil, referring to the present isolated phases with reduced Ni chemical state (Fig. S47c).

## Discussion

How such interphasic synergy promotes the intrinsic activity relates to intermediate interaction with the ready-to-serve surface active sites. Cyclic voltammetry obtained using pristine Ni electrodes at a potential range of −0.05 to 1.3 V, excluding HER and Ni oxidation, depicts stronger anodic current at 0.6 and 1.1 V, assigned to OH$_{ads}$ formation at surface, as well as additional redox couples at 0.85 V for

Ni(OH)$_2$@Ni-N/Ni-C/NF compared to Ni(OH)$_2$/NF (Fig. S48a, b)[49]. The extended intermediate adsorption potential window to 1.2 V, which suggests the formation of surface adsorbed oxygen species (Ni-O$_{ads}$), also depict larger currents for Ni(OH)$_2$@Ni-N/Ni-C/NF. This remarks an enhanced oxophilic character and a higher affinity to OH adsorption by nano-hetero-interfacing of Ni-N/Ni-C phase. The larger OH$_{ads}$ peak over Ni(OH)$_2$@Ni-N/Ni-C/NF eventually weakens to a similar intensity observed on Ni(OH)$_2$/NF as the adsorption of OH charge carriers in the alkaline electrolyte diminishes gradually. The monophasic Ni(OH)$_2$ has also a strong inherent affinity to OH adsorption but with a slow rate, and thus, the OH$_{ads}$ peak remains intact.

Intriguingly, OH$_{ads}$ formation shifts to lower potentials over Ni(OH)$_2$@Ni-N/Ni-C/NF, after reaction under HER potentials (Fig. S15), where the shoulder peak at 0.85 V becomes prominent (Fig. S48c). The sluggish OH adsorption over Ni(OH)$_2$ surface fully retains the OH$_{ads}$ peak after HER, while the accelerated HER kinetics over Ni(OH)$_2$@Ni-N/Ni-C/NF surface includes OH adsorption/desorption which results in further weakening of OH adsorption from OH charge carriers in the alkaline electrolyte during CV cycles. Thus, a slightly smaller OH$_{ads}$ peak is observed at 0.85 V over Ni(OH)$_2$@Ni-N/Ni-C/NF than the peak at 1.1 V on Ni(OH)$_2$/NF. Additional Ni ions during the formation of the nano-heterostructure augments the reversing of OH$_{ads}$ potential. Therefore, although H$_2$ recombination at Heyrovsky step is dominated by the faster H adsorption-desorption at Volmer step, Ni-N/Ni-C phase favors OH adsorption and its synergy with Ni(OH)$_2$ also facilitates OH adsorption-desorption pathways.

The mechanism of the interphasic synergy is scrutinized by ab initio total adsorption energies of *H intermediate via density functional theory (DFT) calculations. The optimum *H adsorption energy is predicted by interfacing Ni(OH)$_2$ with Ni-N/Ni-C phase with two Ni-N coordination, i.e., NiN$_3$-C and NiN$_4$-C (Fig. 5a and Fig. S49). Both Ni-N coordination values have been reported for Ni-N-C active phase in electrocatalysts[33,50,51]. Ni(OH)$_2$@NiN$_3$-C structure has a similar surface energy to Ni(OH)$_2$@NiN$_4$-C assuming the top Ni site for hydrogen adsorption, while both are inferior to the hollow sites in monophasic Ni(OH)$_2$ (Fig. S50). Calculations of the Gibbs adsorption free energies with two different adsorption sites for H$_{ads}$ (ΔG(*H)) gives 1.23 and 0.49 eV for *H adsorption on Ni top site and Ni bridge site in NiN$_3$-C phase in Ni(OH)$_2$@NiN$_3$-C, respectively. Hence, only when a bridge site forged between Ni and C atoms in NiN$_3$-C phase is considered as the adsorption site, a more favorable hydrogen adsorption energy is attained. In addition, the relaxed *OH adsorption on the Ni top site in NiN$_3$-C phase via Gibbs adsorption free energy calculation for OH$_{ads}$ gives a value of 1.744 eV. The loss and gain of Bader charges are calculated during the adsorption of *H, *OH, and H$_2$O on the Ni active site in NiN$_3$-C phase in Ni(OH)$_2$@NiN$_3$-C (Fig. S51), elucidating that the surface interaction of H$_{ads}$ and OH$_{ads}$ adsorbates proceeds preferably at bridge (Ni-C) and top (Ni in NiN$_3$-C) sites, respectively (Fig. 5b, c). The adsorption of *H intermediate on the Ni top and bridge sites, *OH intermediate on the Ni top site, as well as the co-adsorption of dissociated *H-*OH on the Ni top and bridge sites, in NiN$_3$-C phase is modelled over Ni(OH)$_2$@NiN$_3$-C (Fig. S52).

The Gibbs free energy diagram of *H adsorption predicts NiN$_3$-C phase with Ni bridge sites in Ni(OH)$_2$@NiN$_3$-C heterostructure is more favorable at Volmer step and interfacing with Ni(OH)$_2$ renders a smaller hydrogen adsorption energy (Fig. 5d). The adsorption of OH requires an appreciable amount of surface energy (1.74 eV), confirming the oxyphilic nature of top Ni sites in NiN$_3$-C phase. The Gibbs free energy difference slightly increases upon the co-adsorption of *H and *OH species (1.87 eV). Thus, the faster hydrogen adsorption-desorption-recombination process is the main contributor to the accelerated kinetics. Both *H and *OH are adsorbed on top Ni sites in monophasic Ni(OH)$_2$, while the strong *H adsorption increases the resistance for the subsequent H$_2$ recombination.

The projected density of states (PDOS) analysis demonstrates five equivalent energy states (spin-up and down) for Ni(OH)$_2$@NiN$_3$-C structure as well as a shifted $\varepsilon_d$ to low-energy level (−1.98 eV) for surface Ni sites (Fig. 5e). The shift in $\varepsilon_d$ is closer to that of theoretical NiN$_3$-C phase (−1.86 eV) compared with Ni(OH)$_2$ (−2.49 eV) (Fig. S53a, b). This alludes to a large DOS of Ni atoms occupying at Fermi level, which lower the antibonding energy states and greatly contribute to intermediate adsorption. Bader charge analysis demonstrates that the electron gain by *H intermediate is less (0.167 e) on the bridge Ni-C site in Ni(OH)$_2$@NiN$_3$-C than on the top Ni site in Ni(OH)$_2$ (0.590 e). The *H adsorption energy is also inversely proportional to the absolute value of $d$-band center. Hence, the less electron number for *H intermediate and higher number of $d$-states of Ni atoms at Fermi level enhance the intrinsic activity (Fig. 5f).

Comparison of Bader charge variation over monophasic surface with that on Ni(OH)$_2$@NiN$_3$-C reveals the electron loss of Ni atoms in Ni(OH)$_2$ phase and electron gain of C atoms in NiN$_3$-C phase (Fig. 5g). This signifies the enhanced interphasic charge transport from Ni(OH)$_2$ to NiN$_3$-C phase, which not only promotes *H adsorption-desorption but also expedites the desorption of the remaining *OH intermediates. Thus, the co-adsorption of *H and *OH is improved, where the *H adsorbed on the C atom in the bridge Ni-C site accelerates *OH adsorption on the top Ni sites (Fig. S54). The projected charge density difference (CDD) illustrates this via charge redistribution directed by the enhanced hybridization mainly from Ni(OH)$_2$ to NiN$_3$-C through OH groups, as 3$d$ orbitals of Ni atoms in both phases are localized near Fermi level (Fig. 5h). The marked electron accumulation at the interface is responsible for the emergence of interphasic synergy.

In summary, monometallic interphasic synergy is demonstrated between Ni(OH)$_2$ and Ni-N/Ni-C phases for high-performance alkaline HER. Ni nano-hetero-interfaces occur locally in large number at surface and are stable until very large HER current densities. Such nano-hetero-interfacing exerts an interphasic synergy which augments charge redistribution between the two Ni phases, enhances energy level of Ni $d$-state orbitals by upshifting $d$-band center, accelerates hydrogen adsorption-desorption-recombination process by lowering the electron transfer number of *H intermediate, and facilitates *OH adsorption-desorption. This results in ready-to-serve surface active sites that significantly promote interfacial charge transfer kinetics and long-term stability. Elucidation of interphasic synergy in this study sheds light on the tentative concept of electrocatalytic synergy and offers new insights to exploit synergistic interactions between different phases in nonprecious metal-based catalyst structures. Together with intermetallic synergy between certain transition metals, this form of electrocatalytic synergy can elucidate the path for the development of highly efficient electrocatalysts for a range of energy applications.

## Methods
### Materials
Nickel foam (NF, 1.5 mm thickness, 110 ppi) was used as the starting material. Ethylenediamine (EDA), diethylamine (DEA), N,N-diethylmethylamine (DEMA), diethylenetriamine (DETA), 1,1,3,3-Tetramethylguanidine (TMG), formamide (FA), 1-methylimidazole (MI), nickel nitrate hexahydrate (Ni(NO$_3$)$_2$.6H$_2$O), nickel sulfate anhydrous (NiSO$_4$), Ni phthalocyanine (Ni-pc), standard nickel hydroxide in nanoparticle form (Ni(OH)$_2$*), boric acid (H$_3$BO$_3$), nitric acid (HNO$_3$), and ethanol were purchased from Sigma. Milli-Q-water was used for all of solution preparations.

### Preparation of Ni(OH)$_2$@Ni-N/Ni-C/NF and Ni(OH)$_2$/NF
Ni foam was initially acid treated with HNO$_3$ (5 M) in an ultrasonic bath for 15 min to cleanse the oxide layers & impurities, then washed thoroughly with water and ethanol, and kept dried at vacuum oven prior to use. For the fabrication of Ni hetero-hierarchical nanostructure, a piece

of clean NF ($1 \times 3$ cm$^2$) was placed in a homogeneous mixture of water and SDA (EDA, DEA, DEMA, DETA, TMG, FA, and MI) with varying volume ratio (v:v %) with or without trace amounts of Ni(NO$_3$)$_2$.6H$_2$O. The obtained transparent solutions with EDA, DEA, DEMA, DETA, TMG, FA, and MI as SDAs turn into light pink, light red, pale orange, light red, blue, yellow, and reddish-brown colours, respectively by adding trace amounts of Ni(NO$_3$)$_2$.6H$_2$O. Five mL of the resultant solutions was transferred to a 20 mL stainless steel autoclave and kept under various times and temperatures of the solvothermal reaction. The solvothermal-reacted NF was then rinsed with copious water and ethanol and dried in vacuum. Then, it was treated by post-thermal annealing in nitrogen atmosphere at a mild temperature of 400 °C for 2 h. The resultant NF was kept under vacuum before use. When compared, the obtained electrodes are denoted Ni(OH)$_2$@Ni-N/Ni-C/NF$_p$ and Ni(OH)$_2$@Ni-N/Ni-C/NF$_i$ without (pristine NF) and with additional Ni ions in the solvothermal bath. The preparation of monophasic Ni(OH)$_2$/NF electrode was the same except the addition of SDA into the solvothermal bath. Similarly, the monophasic Ni electrodes are denoted as Ni(OH)$_2$/NF$_p$ and Ni(OH)$_2$/NF$_i$ without and with using additional Ni ions when compared directly. The hetero-hierarchical nanostructure obtained with optimal concentration of external Ni ions, demonstrated in Figs. 1–4 and various instances in the supplementary information, is simply presented as Ni(OH)$_2$@Ni-N/Ni-C/NF.

### X-ray absorption spectroscopy (XAS)

Ni K-edge spectra were recorded at XAS beamline, Australian Synchrotron, via fast-freezing of Ni catalysts grown on carbon fiber paper as the inert substrate to avoid the strong self-absorption by metallic Ni and the phase transformation induced by from photo-damage. After performing chronopotentiometric tests at different HER potentials, the electrodes were immediately put into liquid nitrogen. Fast-quenched samples were wrapped with Kapton tape and mounted on cryostat sticks and inserted in a helium-filled cryostat chamber (20 K). The fluorescence data was collected at 45 degrees relative to the incident beam. The X-ray absorption near edge structure (XANES) spectra were calibrated against the first inflection point of Ni foil (8333.0 eV). The collected fluorescence data were initially read into Sakura, then deglitched and processed for background subtraction using a combination of PySpline and Athena[52]. The k-range and Fourier transform at R-space of the extended X-ray absorption fine structure (EXAFS) were processed using Athena. Artemis was used for EXAFS fitting. Quasi-operando XAS was carried out by fast-frozen electrodes immediately quenched in liquid nitrogen after chronoamperometry tests under different HER potentials in 1.0 M KOH using a 760 CHI potentiostat workstation, with Ag/AgCl and Pt wire as reference and counter electrode, respectively.

### Electrochemical measurements

Electrochemical tests were performed in 1.0 M KOH solution in a standard three-electrode system using CHI 760D Electrochemical Workstations (CHI instruments). The as-prepared Ni(OH)$_2$@Ni-N/Ni-C/NF and Ni(OH)$_2$/NF electrodes were directly used for HER with a fixed working area of 0.25 cm$^2$. Mercury-mercury oxide (Hg/HgO) and graphite plate were used as reference and counter electrodes, respectively. Saturated calomel electrode (SCE) was used to test HER in H$_2$SO$_4$ solution. The recorded potentials were converted to reversible hydrogen electrode (RHE) using the following equations.

$$E_{RHE} = E_{Hg|HgO} + 0.140\,V + 0.0592\,pH \qquad (1)$$

$$E_{RHE} = E_{SCE} + 0.241\,V + 0.0592\,pH \qquad (2)$$

The HER polarization was conducted by linear sweep voltammetry (LSV) at a scan rate of 5 mV s$^{-1}$ with an ohmic resistance compensation level of 90% of the total solution resistance. Cyclic voltammetry (CV) was performed at scan rates of 1 and 10 mV s$^{-1}$. Stability test was performed by chronopotentiometry technique. Electrochemical impedance spectroscopy (EIS) was conducted at 50 to 500 mV and a scanning frequency range between 100 kHz to 0.01 Hz at an amplitude of 0.01 V. The Gaussian-basis function DRT of EIS spectra was calculated using an extended version of MATLAB® application DRTtools[45]. The relationship between the measured impedance and DRT is given by[44]:

$$Z(\omega) = R_\infty + R_{pol} \int_{-\infty}^{\infty} \frac{\Gamma(\log(\tau))}{1 + i\omega\tau} d(\log(\tau)) \qquad (3)$$

where Z is impedance, R$_\infty$ is series resistance, R$_{pol}$ is total polarization resistance, $\Gamma$ is DRT, $\tau$ is the relaxation time, and $\omega$ is angular frequency. The DRT parameters were fixed at a regularization parameter of 10$^{-3}$. The deconvolution of individual DRT peaks along with peak maximum frequency identification and area under peak integration was achieved using the MATLAB® software DCscript using Voigt deconvolution.

## Data availability

The supporting data in this work are presented in the paper and the Supplementary Information. Source data are available from the corresponding author upon request.

## Code availability

Code for deconvoluting and analysing the DRT data is available at: https://au.mathworks.com/matlabcentral/fileexchange/52321-peak-fitting-to-either-voigt-or-lognormal-line-shapes[53]

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

## Acknowledgements

The authors thank UNSW Mark Wainwright Analytical Centre (MWAC) including the Electron Microscope Unit (EMU) and Solid State and Elemental Analysis Unit (SSEAU) at UNSW, Sydney, for the access to all characterization instruments, Dr Xianjue Chen for assisting in taking FIB-STEM images, and Mr Shuhao Wang for helping with Bader charges analysis. We also thank Australian Synchrotron and Dr Bernt Johannessen for assisting with the merit XAS beamline. The study was supported by Australian Research Council (FT170100224, DP210103892, IC200100023). X. Shen thanks the National Natural Science Foundation of China (Grant No. 21873086) and the computational times supported by Henan Supercomputer Center in Zhengzhou.

## Author contributions

K.D. designed and carried out the experiments, collected data, and wrote the manuscript. X.S. performed DFT simulation. R.K.H. and K.D. carried out and analysed XAS data. Q.M. performed RSM modelling. K.D. and Q.M. collected, analysed, and interpreted EIS and DRT data. C.Z. supervised the work and co-wrote the manuscript. All authors discussed the results and commented on the manuscript.

## Competing interests

The authors declare no competing interests.

## Additional information

**Peer review information** : *Nature Communications* thanks Ronen Bar-Ziv and the other, anonymous, reviewer(s) for their contribution to the peer review of this work. Peer reviewer reports are available.

