## [Peer Review File · Nature Communications]

Peer Review comments, first round review -

Reviewer #1 (Remarks to the Author):

This work reports a novel interphasic synergy as activity enhancement mechanism for hydrogen evolution reaction (HER) in alkaline electrolytes. The authors demonstrate this in a Ni monometallic heterostructure, and discuss the emergence through hetero-interfacing of two Ni phases of Ni(OH)₂, as a good water dissociator but weak for HER pathways, and Ni-N/Ni-C with suitable surface energy for the adsorption and desorption of H and OH intermediates in a Volmer-Heyrovsky pathway. The authors carried out a systematic study of interphasic synergy using an array of electrochemical, spectroscopic, and operando methods. It is discovered that hetero-interfacing the Ni(OH)₂ and Ni-N/Ni-C phases introduce ready-to-serve Ni active sites at the beginning of Volmer step and tune Ni sites in the heterostructure electronically. Ni is a widely used nonprecious metal for HER in a wide range of catalyst materials, thus understanding the factors of intrinsic activity, long term stability, and structure-activity relationship is important for catalyst development. Overall, this is an interesting and in-depth study and would be useful for electrocatalysis and wide community of various electrochemical applications. I recommend the acceptance of this manuscript after improving the following points.

1- What is the role of amorphous carbon shell surrounding Ni(OH)₂ and Ni-N/Ni-C phases in the heterostructure in the emergence of interphasic synergy and HER performance? The authors did not mention this.

2- Why is no N detected by HRTEM? Do other TEM images of other parts of the heterostructure show Ni-N?

3- The authors should point out to why nano-hetero-interfacing of Ni(OH)₂ and Ni-N/Ni-C phases causes the shift to lower energy in the Ni K-edge in the XANES of the heterostructure compared to the monophasic Ni(OH)₂.

4- What is the reason for the increasing noise in the EXAFS of the heterostructure at larger k values compared to the monophasic Ni(OH)₂?

5- How does the morphology look like at larger HER overpotentials after stability? Are the surface hierarchies preserved?

6- Figure S29 show a well-resolved Raman spectrum for the heterostructure via RSM optimization,

particularly regarding Ni-O and graphitic carbon vibrations. Then, how does the non-optimal structure perform for HER in comparison?

7- How would the trends for $R_{eff,CT}$ and $R_{eff,MT}$ parameters in figure S31 be for the other systems shown with Nyquist plots in figure S30? Also, the authors ascribe the prominent low frequency peak in the DRT plots derived from operando EIS to the resistance of the surface to post-diffusion of reaction intermediates. What is the reason for the appearance of this peak and how such conclusion is made?

8- Figure S37 show increased metallic Ni after HER at an overpotential of 100 mV but the corresponding XANES shows an increase in the photon energy of the Ni K-edge at this overpotential. The reason for this difference should be explained which is missing in the manuscript. Also, unlike the represented XANES after HER at smaller overpotentials in figure S41, the XANES after HER at an overpotential of 600 mV actually shows subtle differences with the linear combination of the as-prepared heterostructure and metallic Ni. What causes the variation in the Ni K-edge at large HER potentials and how different the catalyst structure is compared to the structures suggested from LCA plots at smaller HER potentials?

9- What causes the decrease in the $\Delta\eta/\Delta\log|j|$ function at large current densities for Ni(OH)₂@Ni-N/Ni-C/NF compared to the increasing values for Ni(OH)₂/NF? Also, why the activity loss is bigger for the stability performance at the lower HER current density of -100 mA cm⁻² against that of -500 mA cm⁻²? In the HER polarization curves, how does the shown activity of the benchmark Pt/C catalyst compared to standard data reported in the literature? Is the benchmark Pt/C/NF close to the best Pt electrodes for alkaline HER?

10- More recent literatures are suggested to be referred: Adv. Mater. 2017, 29(2), 1602441; J. Am. Chem. Soc. 2014, 136(21), 7587-7590.

Reviewer #2 (Remarks to the Author):

The manuscript focusses on the interphasic synergy between Ni(OH)₂ and Ni-N/Ni-C phases for hydrogen evolution reaction (HER) in alkaline media. The synthesized material was carefully characterized. The electrocatalytic activity and stability was tested for the application reaction. Further, the impact of nano-hetero-interfacing on interfacial charge transfer between catalyst and electrolyte was investigated by in situ electrochemical impedance spectroscopy. Density functional theory calculations were applied in addition to experimental work, which demonstrates that the interphasic synergy yields ready-to-serve Ni active sites and maintains the promoted intrinsic activity under a wide range of HER potentials.

Generally, the subject of this work is of current interest, as especially proven by numerous recent publications. Acquired data can be applied for the development of highly efficient noble-metal-free electrocatalysts for an energy application. Therefore, the manuscript is suitable for publication in Nature Communications. However, further proofreading is required and the following comments should be addressed:

1. Fig.2 d presents data with seemingly identical upper and lower axis scaling. It is suggested removing the upper red colored scale or (less preferred) considering the addition of an axis caption (red) for the upper axis.

2. For the sake of completeness, authors should define the intensity ratio ID/IG used on page 10: Mention assigned wave numbers and add a hint that band intensities are concerned.

3. The sentence "Operando XAS also reveals a slight decrease in the peak II to III intensity ratio(Figure S25) up to 200 mV, followed by a sharp decline at 600 mV." on page 14 is confusing as fig. S25 presents data on different samples and not data on the effect of reduction at different overpotential. Could authors check if referring to fig. 4i is not intended?

4. The sentence "The hetero-hierarchical nanostructure obtained with optimal concentration of external Ni ions, demonstrated in Figures 1-6, is simply presented as Ni(OH)₂@Ni-N/Ni-C/NF." on page 19 refers to fig.6 despite the fact the only 5 figures are provided.

Reviewer #3 (Remarks to the Author):

[see next page]

This paper presents a very interesting experimental and modeling study of interphasic synergy in monometallic structures (between Ni(OH)₂ and Ni-N/Ni-C phases) for high-performance of HER electrocatalysis in alkaline media. Overall the manuscript is of high quality and could be published in Nature Communications after addressing several issues as follows:

1. From the mechanistic point of view and to draw the complete picture, it is essential to know the effect of pH on HER rate as the kinetics are determined through a subtle balance between the water dissociation (Volmer step) and the subsequent chemisorption of the water-dissociation intermediates (*OH and H*) on the surface of the catalyst. Similar to the HER pathway in an alkaline medium, the acidic HER also occurs via Volmer-Tafel or Volmer-Heyrovsky paths, but the hydrogen intermediate (H*) following the Volmer step is formed by an electron transfer step through the discharge of protons (acid), i.e., $H^+ + e^- \rightarrow H^*$, therefore the hydrogen adsorption energy (ΔG_{H^*}) is the primary descriptor, while in alkaline medium also the energy barrier for water dissociation H--OH, and the OH* interaction is playing a major role. As Ni(OH)₂ is recognized as a good water dissociation, but with too strong H* affinity, as the author also mentioned, I would expect that the differences in performance in acidic media will be even more pronounced compared to Ni(OH)₂@Ni-N/NiC catalyst due to its accelerated hydrogen adsorption-desorption recombination process as the author concluded. Please provide the polarization curves in acidic solutions and elaborate.
2. Parts of the article are not clearly accessible to the reader and make it difficult for the reader to follow and draw conclusions:
 - a. lines 296-297 which refer to **Figure S45**, a detailed legend should be provided (the symbols and the location of the H* and *OH species).
 - b. The authors conclude that the surface interaction of dissociated H_{ads} and OH_{ads} proceeds preferably at bridge (Ni-C) and top (Ni in NiN₃-C) sites, respectively (line 298-299). Where is the calculation that supports this conclusion? Figure 5b and c present only a visual illustration.

Moreover, in the DFT calculations (pages 5-6, SI) the calculations of the hydrogen adsorption energy H^* ($\Delta G(H^*)$) is provided without any details for OH^* adsorption and the energy barrier for water dissociation $H-OH$ calculations.

3. Lines 302-304: “The adsorption of OH requires an appreciable amount of surface energy (1.74 eV)..... The Gibbs free energy difference slightly increases upon the co-adsorption of H^* and OH^* species (1.87 eV)” please support these values with the corresponding figure and calculations.
4. I agree that the strong H^* adsorption on the monophasic $Ni(OH)_2$ increases the resistance for the subsequent H_2 recombination, however, the H^* adsorption free energy (Figure 5) shows that $\Delta G(H^*) = 0.493$ eV for $Ni(OH)_2/NiN_3C$ (bridge), is still not optimal, *i.e.* far from thermoneutral hydrogen binding energy ($G \sim 0$), please discuss this point.
5. (a) In Fig S42 and lines 278-283, - the author claimed that the OH_{ads} formation potentials shift to lower potential after HER. If this is the case, I'd expect for changes in the HER LSV curves after 1st, 5th, 10th ... cycles. Please clarify this point.
(b) In addition, in Fig S42c, "after HER" please explain? conditions, potentials ? number of scans ?
(c) Fig S42 (a-b) depicts a larger anodic current density at ≈ 1.1 V for $Ni(OH)_2@Ni-N/Ni-C/NF$ compared to $Ni(OH)_2$, however, a comparison of CV curves after HER (S42c) shows the opposite, with a prominent peak for $Ni(OH)_2$ at 1.1 V(Fig S42c). Please discuss.
6. Fig S13(b) should be with ohmic resistance compensation (iR correction). Please check.

Reviewer #1

This work reports a novel interphasic synergy as activity enhancement mechanism for hydrogen evolution reaction (HER) in alkaline electrolytes. The authors demonstrate this in a Ni monometallic heterostructure, and discuss the emergence through hetero-interfacing of two Ni phases of Ni(OH)₂, as a good water dissociator but weak for HER pathways, and Ni-N/Ni-C with suitable surface energy for the adsorption and desorption of H and OH intermediates in a Volmer-Heyrovsky pathway. The authors carried out a systematic study of interphasic synergy using an array of electrochemical, spectroscopic, and operando methods. It is discovered that hetero-interfacing the Ni(OH)₂ and Ni-N/Ni-C phases introduce ready-to-serve Ni active sites at the beginning of Volmer step and tune Ni sites in the heterostructure electronically. Ni is a widely used nonprecious metal for HER in a wide range of catalyst materials, thus understanding the factors of intrinsic activity, long term stability, and structure-activity relationship is important for catalyst development. Overall, this is an interesting and in-depth study and would be useful for electrocatalysis and wide community of various electrochemical applications. I recommend the acceptance of this manuscript after improving the following points.

1- What is the role of amorphous carbon shell surrounding Ni(OH)₂ and Ni-N/Ni-C phases in the heterostructure in the emergence of interphasic synergy and HER performance? The authors did not mention this.

Response. According to previous reports, both pure crystalline and pure amorphous catalysts lack fast interfacial charge transport between catalyst and electrolyte during water splitting reactions¹. Amorphous phases, especially carbon, are reported to contain unsaturated surface sites and endow structural flexibility through randomly-oriented bonding, which facilitate the adsorption of intermediates and the associated charge transport²⁻⁴. For example, enhanced electronic conductivity, stability of surface active sites, and improved HER performance of Ni₂P catalyst was reported by the formation of a carbon shell rich with defects and disordered sites which promoted π -bonding and electron donor-acceptor property in Ni₂P@amorphous carbon composite.⁵ Furthermore, amorphous carbon is thermodynamically more stable than metallic oxides and metal non-oxides^{6,7}. Ni(OH)₂@Ni-N/Ni-C in our study has weak crystalline and short order Ni phases randomly embedded in an amorphous carbon framework throughout the surface. The formation of the amorphous carbon layer promotes nano-hetero-interfacing by encasing the Ni phases at short range order and stabilizing them in alkaline medium, and thus improves interphasic synergy between the Ni phases for HER. Furthermore, the amorphous carbon framework boosts the electronic conductivity of the nano-heterostructure, hence contributes to accelerating the reaction rate at kinetically- and mass transport-controlled regions and the emergence of ready-to-serve Ni active sites, as elucidated by HER kinetics and operando EIS.

Regarding the importance of the amorphous carbon layer, the following is added to the revised manuscript, pages 5-6, lines 103-105:

The formation of the amorphous carbon layer promotes nano-hetero-interfacing by encasing and stabilizing Ni structures in the vicinity and at short-range order as well as by boosting the electronic conductivity.

2- Why is no N detected by HRTEM? Do other TEM images of other parts of the heterostructure show Ni-N?

Response. We are grateful to the reviewer's attention to this detail. A combined and systematic analysis by physical characterization of the electronic structure of the nano-heterostructure as well as DFT simulations elucidate coordinated Ni-N and Ni-C microenvironments on the surface. In fact, N atoms coordinated to Ni and C (as in NiN₃C phase) are crucial for the optimal surface energy for water dissociation and intermediate adsorption-desorption mechanisms. Furthermore, HAADF-STEM-EDS elemental mapping depict the increase in the distribution of O, N, and C atoms near the surface (Figure 1e). Thus, no lattice spacing due to Ni-N-C phase and no individual crystallinity due to a Ni nitride structure (e.g., Ni_xN) could be formed in the Ni(OH)₂@Ni-N/Ni-C nano-heterostructure to be detected by HRTEM. Instead, N atoms are coordinated with Ni and C atoms in Ni-N/Ni-C phases. A similar conclusion was reported by Lei et al³ with a Ni nanoparticles@Ni-N-C heterostructure synthesized by reacting Ni precursors with dicyanamide as the precursor for C and N atoms. The authors reported embedded Ni nanoparticles in a graphitic carbon framework with atomically dispersed Ni-N_x species. They also presented a HAADF-STEM-EDS elemental mapping profile of the heterostructure with uniformly distributed Ni, N, and C elements, but no lattice fringes due to crystallization of Ni_xN structures. This again refers to the formation of atomically coordinated N atoms to Ni atoms in Ni-N-C phases in the reported heterostructure.

3- The authors should point out to why nano-hetero-interfacing of Ni(OH)₂ and Ni-N/Ni-C phases causes the shift to lower energy in the Ni K-edge in the XANES of the heterostructure compared to the monophasic Ni(OH)₂.

Response. The observed shift to lower energy in Ni k-edge in the XANES of Ni(OH)₂@Ni-N/Ni-C nano-heterostructure is associated with a change in effective nuclear charge. Particularly, the linear combination of metallic Ni and Ni(OH)₂ did not explain the shift, indicating it is associated with a reduction in charge. The observed shift, thus, can be explained by two main reasons:

1) The nano-hetero-interfacing of Ni(OH)₂ and Ni-N/Ni-C phases damps and distorts Ni bonding and endows electron accepting feature to the surface, thus altering the electronic structure in the nano-heterostructure. This systematically investigated by XAS, where the damped and broadened conjunct Ni-O/Ni-N scattering pair (Figure 3c) as well as the appearance of Ni-Ni scattering pair between those of Ni foil and Ni(OH)₂ (Figure 3d) indicate the altered Ni coordination environment at the local interface between Ni(OH)₂ and Ni-N/Ni-C phases and the endowed metallicity to the electronic structure. The boosted charge transport between Ni(OH)₂ and Ni-N/Ni-C phases entails distinctive differences in the XANES of the nano-heterostructure with the linear combination of metallic Ni and Ni(OH)₂ (Figure S27a), involving the reduced chemical states of Ni centres.

2) The interphasic charge transport is projected from Ni(OH)₂ to Ni-N/Ni-C moieties as a result of significantly increased d-states of Ni atoms close to Fermi level as well as the Bader charge variation with electron loss at Ni(OH)₂ phase and electron gain at NiN₃-C phase. Such interphasic charge transport is verified by the upshift of the valence band to Fermi level (Figure 3e and f), which renders electron accumulation at the local interface of Ni phases (Figure 5h), increasing the electron density on surface Ni sites and reducing the oxidation state.

As a result, the enhanced electron density and endowed metallicity shift the Ni K-edge to lower energy in the XANES spectrum of Ni(OH)₂@Ni-N/Ni-C. This is also confirmed by the shift in Ni 2p XPS spectrum of Ni(OH)₂@Ni-N/Ni-C to lower binding energy compared to that of Ni(OH)₂ (**Figure R1**).

Figure R1. The shift of Ni 2p XPS peaks to lower binding energies in Ni(OH)₂@Ni-N/Ni-C compared to Ni 2p XPS peaks in the monophasic Ni(OH)₂.

The change in the chemical state of Ni sites in the nano-heterostructure according to the above reasons is added to the revised manuscript, page 11, lines 214-219, and **Figure R1** is added to the revised Supplementary Information as Figure S32.

Similarly to the shift of Ni K-edge in the XANES of Ni(OH)₂@Ni-N/Ni-C to lower energy compared to the monophasic Ni(OH)₂ structure, the Ni 2p features in the corresponding XPS spectrum also shifts to lower binding energy (**Figure S32**). The observed shifts to lower energy allude to the enhanced electron density and endowed metallicity because of the altered Ni coordination environment and boosted interphasic charge transport at the local interface between Ni(OH)₂ and Ni-N/Ni-C phases.

4- What is the reason for the increasing noise in the EXAFS of the heterostructure at larger k values compared to the monophasic Ni(OH)₂?

Response. We thank the reviewer for the scrutiny. Indeed, almost all nano-heterostructure samples depict disarranged EXAFS at larger K values and the noise level increases for the samples subjected to applied potentials. Experimentally, this could be attributed to the enhanced distortion of Ni bonding and coordination environments via nano-hetero-interfacing of Ni(OH)₂ and Ni-N/Ni-C phases. As elucidated by the broadened and damped EXAFS in K and R space as well as the enhanced charge re-distribution from Ni(OH)₂ to Ni-N/Ni-C phase by DFT calculations, a large amount of Ni 3d states localize at the interface of the two Ni phases. This brings about higher electron delocalization during XAS measurement which elevate the noise level. It should be also noted that given the weak crystallinity of the Ni(OH)₂@Ni-N/Ni-C nano-heterostructure (Figure S12) and amorphous carbon layer at the surface, the XANES and EXAFS intensities are damped which decrease the signal to noise ratio compared to the sharp XANES and EXAFS of monophasic Ni(OH)₂. Also, mathematically, the noise level in EXAFS profiles increases when the EXAFS intensity decreases. This generally happens when two or more components are present or there is disorder in a system.

5- How does the morphology look like at larger HER overpotentials after stability? Are the surface hierarchies preserved?

Response. We thank the reviewer for the comment. We acquired SEM images of Ni(OH)₂@Ni-N/Ni-C/NF after long-term stability test at -500 mA cm⁻² to investigate the morphology and surface hierarchies, following the reviewer's comment. We found out overall the hierarchical morphology with surface hierarchies grown at anisotropic architectural zones with different growth orientations are preserved even at larger current densities (equivalent to larger potentials) (**Figure R2**).

Figure R2. SEM micrographs of Ni(OH)₂@Ni-N/Ni-C/NF during HER at -500 mA cm⁻².

To address this point in the study, **Figure R2** is added to the revised Supplementary Information file as Figure S20, and the following is added to the revised manuscript, page 8, lines 145-146:

The surface hierarchies are preserved after 100 h of HER at -100 and -500 mA cm⁻² (**Figure S19 and S20**).

6- Figure S29 show a well-resolved Raman spectrum for the heterostructure via RSM optimization, particularly regarding Ni-O and graphitic carbon vibrations. Then, how does the non-optimal structure perform for HER in comparison?

Response. We thank the reviewer for bringing the point about the optimization methodology to induce nano-hetero-interfacing between Ni(OH)₂ and Ni-N/Ni-C phases and the emergence of interphasic synergy. Similar but much weaker Ni-O and graphitic carbon vibrations detected for the non-optimal heterostructure in the corresponding Raman spectrum in Figure S34b allude to obtaining a non-optimal structure during RSM optimization and an insignificant synergistic interaction between Ni phases. Indeed a weaker HER activity is attained compared to the optimal nano-heterostructure (**Figure R3**).

Figure R3. Comparison of HER LSV curves of the optimal $\text{Ni(OH)}_2@Ni-N/Ni-C$ nano-heterostructure, non-optimal $\text{Ni(OH)}_2@Ni-N/Ni-C$ heterostructure, and monophasic Ni(OH)_2 in 1 M KOH.

The comparison of HER activity between optimal nano-heterostructure and non-optimal heterostructure is added to the revised Supplementary Information as Figure S35. The following is also revised in the revised manuscript in page 12, lines 229-231:

The effect of Ni ion concentration and RSM optimization is indicated by the weakening of two-phonon Ni-O and graphitic carbon vibrations (**Figure S34b**) and the inferior HER performance of the non-optimal heterostructure (**Figure S35**).

7- How would the trends for $R_{\text{eff,CT}}$ and $R_{\text{eff,MT}}$ parameters in figure S31 be for the other systems shown with Nyquist plots in figure S30? Also, the authors ascribe the prominent low frequency peak in the DRT plots derived from operando EIS to the resistance of the surface to post-diffusion of reaction intermediates. What is the reason for the appearance of this peak and how such conclusion is made?

Response. We thank the reviewer for such insightful notice in the interfacial charge and mass transport mechanisms with respect to hetero-interfacing the Ni phases. We calculated R_{CT} and R_{MT} values for $\text{Ni(OH)}_2@Ni-N/Ni-C$ nano-heterostructure and monophasic Ni(OH)_2 without the addition of ionic Ni to the solvothermal reaction, where Ni structures emanate from Ni foam. Consequently, we obtained the trends for $R_{\text{eff,CT}}$ and $R_{\text{eff,MT}}$ parameters for these samples by normalizing R_{CT} and R_{MT} values by the potentials corresponding to a current density of 10 mA cm^{-2} throughout HER active region (**Figure R4**). The obtained trends elucidate a similar impact of nan-hetero-structuring on providing a reactive surface for Faradaic and capacitive processes and the boost in the intrinsic activity of Ni active sites, in the absence of additional ionic Ni source.

To include the $R_{\text{eff,CT}}$ and $R_{\text{eff,MT}}$ trends obtained over $\text{Ni(OH)}_2@Ni-N/Ni-C/NF$ and $\text{Ni(OH)}_2/NF$ derived from Ni foam as Ni source, **Figure R4** is added to the revised Supplementary Information file as Figure S37b.

Figure R4. Comparison of Effective interfacial charge transfer and mass transport of $\text{Ni(OH)}_2@Ni-N/Ni-C/NF$ and $\text{Ni(OH)}_2/NF$.

DRT peak deconvolution denotes time constants corresponding to high, medium, and low frequencies that decouple the interfacial charge transfer process following the HER mechanism, i.e., mass transport from bulk electrolyte to surface active sites (ASR_{MT} at high frequency), reaction intermediate adsorption (ASR_{CT} at medium frequency), and diffusion of the H_2 gas product from the surface as a result of $*H-*H$ intermediate recombination (ASR_{PD} at low frequency). As explained by DFT simulation results, a $*H$ adsorption-desorption-recombination process is the primary activity descriptor for HER kinetics. Thus, the resistance to intermediate desorption and recombination in the form of molecular H_2 product at the final step of HER mechanism corresponds to the deconvoluted ASR_{PD} peak at low frequency in DRT plots. Similar attribution of the peak at low frequency in Bode-phase plot from in situ EIS was made to the formation and desorption of O_2 gas products following the deprotonation of surface $*OOH$ intermediates over defective Co_3O_4 catalysts during the last step of oxygen evolution reaction.⁸ Having interphasic synergy, $\text{Ni(OH)}_2@Ni-N/Ni-C/NF$ does not show ASR_{PD} peak at low overpotentials (<100 mV), referring to the high intrinsic activity of Ni sites at the beginning of HER and the intact hetero-hierarchical nanostructure. The appearance of a small ASR_{PD} peak at 100 mV alludes to the partial disruption of nano-hetero-interfaces during HER, as further verified by operando XAS. In comparison, the dominance of ASR_{PD} over ASR_{CT} in the deconvoluted DRT of monophasic $\text{Ni(OH)}_2/NF$ suggests that $*H$ desorption and recombination to H_2 product is the RDS instead of $*H$ adsorption at the beginning of HER at low potentials. Even by switching the RDS to ASR_{CT} by applying higher potentials, ASR_{PD} remains prominent for $\text{Ni(OH)}_2/NF$, further signifying the less effective interfacial charge transfer even by providing more energy input. This is because the strong H^* adsorption over Ni top sites of Ni(OH)_2 increases the resistance for its subsequent recombination and post-diffusion into H_2 product.

8- Figure S37 show increased metallic Ni after HER at an overpotential of 100 mV but the corresponding XANES shows an increase in the photon energy of the Ni K-edge at this overpotential. The reason for this difference should be explained which is missing in the manuscript. Also, unlike the represented XANES after HER at smaller overpotentials in figure S41, the XANES after HER at an overpotential of 600 mV actually shows subtle differences with the linear combination of the as-prepared heterostructure and metallic Ni. What causes

the variation in the Ni K-edge at large HER potentials and how different the catalyst structure is compared to the structures suggested from LCA plots at smaller HER potentials?

Response. We thank the reviewer for bringing up this point. XPS was taken on Ni(OH)₂@Ni-N/Ni-C/NF and Ni(OH)₂/NF electrodes while XAS was performed on Ni(OH)₂@Ni-N/Ni-C/Ni/CFP and Ni(OH)₂/Ni/CFP electrodes to avoid the strong self-absorption by Ni foam. The reason for the upshift of the XANES edge at the overpotential of 100 mV, as discussed in the manuscript, is the partial disruption of Ni(OH)₂ and Ni-N/Ni-C phases which is supported by DRT deconvolution at the same overpotential (appearance of ASR_{PD} peak at 100 mV, Figure 4d) and the shift of the second deconvoluted Ni 2p_{3/2} XPS peak to higher binding energy (Figure S43). As a result of size decrease of segregated Ni-N/Ni-C moieties, partial oxidation of Ni atoms occurs. Whereas the increase in the intensity of Ni⁰ peaks in the corresponding Ni 2p XPS spectra after HER is due to the partial loss of catalysts at the surface, as indicated by the attenuated Ni 2p_{1/2} and 2p_{3/2} peaks and shakeup satellites, as well as the exposure of the bulk metallic Ni from Ni foam substrate to the incident X-ray beam.

To avoid confusion regarding this matter, the corresponding discussion for the XPS spectra after HER is revised in page 15, lines 285-290, as follows:

Post-HER core-level Ni 2p XPS spectra at 100 mV at different time periods reveal attenuated Ni 2p_{1/2} and 2p_{3/2} peaks and shakeup satellites as well as increased metallic Ni due to partial catalyst loss at surface and the exposure of the bulk metallic Ni. Interestingly, a negligible shift is observed for the deconvoluted Ni 2p_{3/2} at 854.6 eV while a considerable shift to higher binding energy is observed for the deconvoluted Ni 2p_{3/2} at 855.7 eV, alluding to the partial disruption of the Ni phases along with the size decrease of Ni-N/Ni-C moieties (Figure S43).

The major difference between the LCA plots in Figure S47 is that the post-HER XANES spectra of Ni(OH)₂@Ni-N/Ni-C reacted at the overpotentials of 100 and 200 mV is similar to the linear component curve and the XANES spectrum of the as-prepared nano-heterostructure. Whereas, as the reviewer suggests, there are subtle differences between the post-HER XANES spectrum of Ni(OH)₂@Ni-N/Ni-C conducted at 600 mV and the linear component of the nano-heterostructure and Ni foil. This happens because in the case of LCA curves at 100 and 200 mV, only partial phase disruption occurs and to the most part the nano-hetero-interfacing of Ni(OH)₂ and Ni-N/Ni-C phases is preserved during the HER at low and moderate overpotentials. This is already confirmed by the weak changes in both peak II to III intensity ratio and Ni K-edge shift up to 200 mV (Figure 4i). However, the sharp decrease in peak II to III intensity ratio and Ni K-edge shift at 600 mV indicate the considerable separation of the Ni phases at the surface. Hence, both the post-HER XANES at 600 mV and the linear component are significantly different with the component 1 (as-prepared Ni(OH)₂@Ni-N/Ni-C), as the Ni structures reduce at the large overpotential. However, the subtle differences observed between the post-HER XANES at 600 mV and the linear component at the edge, white line, and EXAFS regions indicate that the segregated Ni(OH)₂ and Ni-N/Ni-C phases are still preserved in vicinity of each other at the surface and catalyse HER with the dominance of Ni(OH)₂. In fact, the Ni K-edge EXAFS profile at 600 mV is more similar to those recorded at lower overpotentials than to the EXAFS profile of metallic Ni foil (Figure S41). Therefore, although the catalyst structure is different from the as-prepared nano-heterostructure, where the Ni K-edge and nano-hetero-interfacing are significantly reduced, the structure is not an average combination of metallic Ni, Ni(OH)₂, and Ni-N/Ni-C structures, and the segregated Ni phases exist until large HER potentials.

9- What causes the decrease in the $\Delta\eta/\Delta\text{Log}|j|$ function at large current densities for Ni(OH)₂@Ni-N/Ni-C/NF compared to the increasing values for Ni(OH)₂/NF? Also, why the

activity loss is bigger for the stability performance at the lower HER current density of -100 mA cm^{-2} against that of -500 mA cm^{-2} ? In the HER polarization curves, how does the shown activity of the benchmark Pt/C catalyst compared to standard data reported in the literature? Is the benchmark Pt/C/NF close to the best Pt electrodes for alkaline HER?

Response. We thank the reviewer for the comments. The trend of $\Delta\eta/\Delta\text{Log}|j|$ as a function of current density gives a measure of how fast the reaction rate changes in the polarization curve as higher potentials delivering larger current densities are applied. We plotted the change in overpotential and log of the equivalent current density at a wide range of current density (**Figure R5**). At small current densities up to 100 mA cm^{-2} , the change in $\Delta\text{Log}|j|$ is more prominent, while $\Delta\eta$ becomes influential between $100\text{-}500 \text{ mA cm}^{-2}$, to increase the $\Delta\eta/\Delta\text{Log}|j|$ function for $\text{Ni(OH)}_2@\text{Ni-N/Ni-C/NF}$ (**Figure R5a**). $\Delta\eta$ remains the prominent parameter above 500 mA cm^{-2} , where the sharp decrease in the overpotential difference between 500 and 1000 mA cm^{-2} reduce $\Delta\eta/\Delta\text{Log}|j|$. This implicates the retained intrinsic activity and accelerated HER kinetics at very large current densities. In comparison, both $\Delta\eta$ and $\Delta\text{Log}|j|$ impact the $\Delta\eta/\Delta\text{Log}|j|$ function at all current densities for $\text{Ni(OH)}_2/\text{NF}$, where the increase in $\Delta\eta$ and decrease in $\Delta\text{Log}|j|$ alternatively keep increasing $\Delta\eta/\Delta\text{Log}|j|$ more drastically (**Figure R5b**). This alludes to the deteriorated intrinsic activity and decrease in HER kinetics of monophasic $\text{Ni(OH)}_2/\text{NF}$ with current density.

Figure R5. Comparison of $\Delta\eta$ and $\Delta\text{Log}|j|$ trends as a function of current density for (a) $\text{Ni(OH)}_2@\text{Ni-N/Ni-C/NF}$ and (b) $\text{Ni(OH)}_2/\text{NF}$.

To address this point, the following is added in the revised manuscript, page 7, lines 131-134.

The more prominent decrease in the overpotential difference between the large current densities of 500 and 1000 mA cm^{-2} compared to the impact of the current density difference reduces the $\Delta\eta/\Delta\text{Log}|j|$ ratio, implicating the retained intrinsic activity and accelerated HER kinetics at large current densities for $\text{Ni(OH)}_2@\text{Ni-N/Ni-C/NF}$.

To confirm the observed difference in the long-term stability performances at 100 mA cm^{-2} and larger current densities, we repeated the stability at 500 mA cm^{-2} for $\text{Ni(OH)}_2@\text{Ni-N/Ni-C/NF}$. Similar to Figure 2f, a smaller activity loss is again observed (**Figure R6**).

Figure R6. Repeat of stability performance of Ni(OH)₂@Ni-N/Ni-C/NF at -500 mA cm⁻².

We attribute this difference to the retained surface hierarchies (**Figure R2**), presence of segregated Ni(OH)₂ and Ni-N/Ni-C phases (Figure S47c), and attenuated $\Delta\eta/\Delta\text{Log}|j|$ ratio at large current densities (Figure 2c and **Figure R5a**), resulting in a stable performance. Around 100 mA cm⁻², partial disruption of Ni(OH)₂ and Ni-N/Ni-C phases occur for the first time as described with the observations of the appearance of ASR_{PD} peak in the deconvoluted DRT plots in Figure 4d as well as the shift of the XANES edge to higher energy (Figure 4j) and the second deconvoluted Ni 2p XPS peak to higher binding energy (Figure S43). The results of these operando tests suggest sudden structural change in the nano-heterostructure that occur when current density reaches to 100 mA cm⁻² and include partial oxidation of Ni atoms and the size decrease of Ni-N/Ni-C moieties, as nano-hetero-interfacing is partially disrupted. Although such partial phase disruption is not significant to deteriorate the intrinsic activity of Ni active sites and the emerged interphasic synergy, it causes more activity loss on the pristine nano-heterostructure as it occurs for the first time. Whereas even with less nano-hetero-interfacing at larger current densities, segregated Ni(OH)₂ and Ni-N/Ni-C phases retain the HER performance for considerably long periods.

We further repeated the fabrication of 20% Pt/C/NF electrode according to an established method.⁹ We obtained a higher activity for the Pt/C/NF electrode in 1 M KOH with an onset potential of almost 0 V. The performance of Pt/C/NF electrode is compared with those of the recently reported Pt/C/NF electrodes for HER in alkaline media in **Table R1** below. As can be seen, the overpotentials needed to deliver current densities of 10 and 100 mA cm⁻² are close or better than those reported for Pt/C in the literature.

Table R1. Comparison of the 20% Pt/C catalyst over Ni foam electrode presented in this work for HER in 1 M KOH with those of Pt/C electrodes reported recently in the literature.

Electrode	Onset potential (mV)	η_{10} (mV)	η_{100} (mV)	Reference
Pt/C/NF	~ 0	~ 27	~ 93	This work
Pt/C/FC ^a	~ 0	36	139	10
Pt/C/FC	~ 0	20	-	11
Pt/C/FC	~ 0	28	137	12
Pt/C/TM ^b	-	35	-	13
Pt/C/NF	-	47	~ 190	14
Pt/C/NF	-	46	~ 90	15
Pt/C/NF ^c	~ 0	20	~ 85	16
Pt/C/NF	~ 0	24	-	17
Pt/C/NF	-	-	130	18
Pt/C/NF	-	20	100	19
Pt/C/NF	-	19	110	20
Pt/C/NF	-	-	159	21
Pt/C/NF	-	~ 41	-	22
Pt/C/NF	-	~ 30	~ 90	9

^a Fabric cloth, ^b Ti mesh, ^c 40 wt.% Pt/C,

Therefore, we replaced the performance of the presented Pt/C/NF electrode in the polarization curves (Figure 2a, Figure S13) with that of the newly fabricated Pt/C/NF in the revised manuscript and Supplementary Information file.

10- More recent literatures are suggested to be referred: Adv. Mater. 2017, 29(2), 1602441; J. Am. Chem. Soc. 2014, 136(21), 7587-7590.

Response. We thank the reviewer for the useful recommendation. The references are added to the revised manuscript.

Reviewer #2

The manuscript focusses on the interphasic synergy between Ni(OH)₂ and Ni-N/Ni-C phases for hydrogen evolution reaction (HER) in alkaline media. The synthesized material was carefully characterized. The electrocatalytic activity and stability was tested for the application reaction. Further, the impact of nano-hetero-interfacing on interfacial charge transfer between catalyst and electrolyte was investigated by in situ electrochemical impedance spectroscopy. Density functional theory calculations were applied in addition to experimental work, which demonstrates that the interphasic synergy yields ready-to-serve Ni active sites and maintains the promoted intrinsic activity under a wide range of HER potentials. Generally, the subject of this work is of current interest, as especially proven by numerous recent publications. Acquired data can be applied for the development of highly efficient noble-metal-free electrocatalysts for an energy application. Therefore, the manuscript is suitable for publication in Nature Communications. However, further proofreading is required and the following comments should be addressed:

1. Fig.2 d presents data with seemingly identical upper and lower axis scaling. It is suggested removing the upper red coloured scale or (less preferred) considering the addition of an axis caption (red) for the upper axis.

Response. We thank the reviewer for the suggestion. Accordingly the upper red coloured scale is removed and all curves from left and right current density scales have the same potential window.

2. For the sake of completeness, authors should define the intensity ratio I_D/I_G used on page 10: Mention assigned wave numbers and add a hint that band intensities are concerned.

Response. We thank the reviewer for the recommendation. Accordingly, the I_D/I_G intensity ratio is defined, the D-band and G-band wave numbers are mentioned, and it is directly stated that the band intensities of D- and G-bands are being discussed in the revised manuscript. To provide the required information, the following is added to the revised manuscript in pages 11-12, lines 222-229.

The D-band around 1335 cm⁻¹ and G-band around 1566 cm⁻¹ represent the defective and crystalline graphitic structure, respectively. The intensity ratio of D and G bands, depicted as I_D/I_G , is therefore an indicator of the degree of graphitization, in that low intensity ratios ($I_D/I_G < 1$) reveal a high degree of graphitization while high intensity ratios ($I_D/I_G > 1$) refer to a defective and disordered graphitic structure inclining to amorphicity.⁴¹ Thus, the slightly higher intensity of D band than G band in the optimal nano-heterostructure with added ionic Ni in the solvothermal reaction with an I_D/I_G ratio of 1.12 indicates a defective and amorphous graphitic network, while Ni(OH)₂/NF only depicts broad and weak Ni-O vibrations.

3. The sentence “Operando XAS also reveals a slight decrease in the peak II to III intensity ratio (Figure S25) up to 200 mV, followed by a sharp decline at 600 mV.” on page 14 is confusing as fig. S25 presents data on different samples and not data on the effect of reduction at different overpotential. Could authors check if referring to fig. 4i is not intended?

Response. The reviewer is right and we are sorry for the oversight. The reference in the sentence is now corrected to Figure 4i.

4. The sentence “The hetero-hierarchical nanostructure obtained with optimal concentration of external Ni ions, demonstrated in Figures 1-6, is simply presented as Ni(OH)₂@Ni-N/Ni-C/NF.” on page 19 refers to fig.6 despite the fact the only 5 figures are provided.

Response. The reviewer is right and we are sorry to miss for the oversight. The sentence is revised and the term Figures 1-6 is now corrected to Figures 1-4, as in fact in Figure 5 the simulated Ni(OH)₂@NiN_x-C structure is outlined. Therefore, the above mentioned sentence in the revised manuscript, page 21, lines 425-427, stands out:

The hetero-hierarchical nanostructure obtained with optimal concentration of external Ni ions, demonstrated in Figures 1-4 and various instances in the supplementary information, is simply presented as Ni(OH)₂@Ni-N/Ni-C/NF.

This paper presents a very interesting experimental and modelling study of interphasic synergy in monometallic structures (between Ni(OH)₂ and Ni-N/Ni-C phases) for high-performance of HER electrocatalysis in alkaline media. Overall the manuscript is of high quality and could be published in Nature Communications after addressing several issues as follows:

1. From the mechanistic point of view and to draw the complete picture, it is essential to know the effect of pH on HER rate as the kinetics are determined through a subtle balance between the water dissociation (Volmer step) and the subsequent chemisorption of the water-dissociation intermediates (*OH and H*) on the surface of the catalyst. Similar to the HER pathway in an alkaline medium, the acidic HER also occurs via Volmer-Tafel or Volmer-Heyrovsky paths, but the hydrogen intermediate (H*) following the Volmer step is formed by an electron transfer step through the discharge of protons (acid), i.e., $H^+ + e^- \rightarrow H^*$, therefore the hydrogen adsorption energy (ΔG_{H^*}) is the primary descriptor, while in alkaline medium also the energy barrier for water dissociation H-OH, and the OH* interaction is playing a major role. As Ni(OH)₂ is recognized as a good water dissociation, but with too strong H* affinity, as the author also mentioned, I would expect that the differences in performance in acidic media will be even more pronounced compared to Ni(OH)₂@Ni-N/Ni-C catalyst due to its accelerated hydrogen adsorption/desorption recombination process as the author concluded. Please provide the polarization curves in acidic solutions and elaborate.

Response. We thank the reviewer for bringing this point to our attention. We acknowledge the underlying mechanism would entail further difference between Ni(OH)₂@Ni-N/Ni-C nano-heterostructure and Ni(OH)₂ monophasic catalyst for HER in acidic media. To test this hypothesis, we performed HER in 0.5 M H₂SO₄ solution in a standard three-electrode system with graphite plate and saturated calomel electrode (SCE) as counter and reference electrodes, respectively. Although, the HER performance is not as strong as the presented performance in alkaline electrolyte in this study, which is merely due to the weaker intrinsic activity of Ni as active site compared to benchmark noble metals such as Pt and Ru, a similar trend in activity enhancement is observed due to nano-hetero-interfacing between Ni(OH)₂ and Ni-N/Ni-C phases (**Figure R7a**). In acidic medium, no significant enhancement in HER activity is obtained by Ni(OH)₂ with respect to bare Ni substrate, due to its non-optimal surface energy for H* intermediate desorption, rendering large overpotentials of 208 and 450 mV to deliver current densities of -10 and -100 mA cm⁻² (**Figure R8a**), as well as a large Tafel slope of 260.9 mV dec⁻¹, covering current densities up to 100 mA cm⁻² (**Figure R7b**). In comparison, Ni(OH)₂@Ni-N/Ni-C shows much lower overpotentials of 98 and 247 mV for delivering the same current densities (**Figure R8a**), as well as a much smaller Tafel slope of 136.4 mV dec⁻¹ (**Figure R7b**). The difference in Tafel slopes is particularly larger than the differences in those obtained in alkaline medium. Hence, activity enhancement because of interphasic synergy accelerates HER kinetics in acidic environment faster than in alkaline. In addition, comparing the overpotentials obtained in acid and alkaline media to deliver current densities of -10 and -100 mA cm⁻² by Ni(OH)₂@Ni-N/Ni-C and Ni(OH)₂ catalysts verifies larger overpotential differences caused by nano-hetero-interfacing of Ni phases and the corresponding interphasic synergy in acidic electrolyte (**Figure R8c**).

Figure R7. (a) HER polarization curves of bare Ni foam, Ni(OH)₂/NF, Ni(OH)₂@Ni-N/Ni-C/NF, and Pt/C/NF in 0.5 M H₂SO₄, and (b) the corresponding Tafel slopes.

Figure R8. Comparison of the overpotentials obtained by Ni(OH)₂/NF and Ni(OH)₂@Ni-N/Ni-C/NF to deliver current densities of -10 and -100 mA cm⁻² in (a) acid and (b) alkaline electrolytes. (c) Comparison of the differences in the overpotentials derived by Ni(OH)₂/NF and Ni(OH)₂@Ni-N/Ni-C/NF at current densities of -10 and -100 mA cm⁻² between acidic and alkaline electrolytes.

To address the above observations, **Figures R7** and **R8** are added as Figures S23 and S24 as well as the following in the revised manuscript, page 8, lines 151-164:

Modulating monometallic electronic structures and the underlying interphasic synergy is universal strategy for developing new nonprecious metal-based catalysts. In fact, the nano-hetero-interfacing of Ni(OH)₂, as a good water dissociation promoter but with overly strong affinity for hydrogen, with Ni-N/Ni-C phase entails even more pronounced differences in HER electrocatalysis with monophasic Ni(OH)₂ in acidic media. Hydrogen intermediate following the Volmer step forms through the discharge of protons in acidic environment. Therefore,

accelerating the hydrogen adsorption-desorption-recombination process is the primary activity descriptor at low pH. A similar trend in activity enhancement is observed by Ni(OH)₂@Ni-N/Ni-C in 0.5 M H₂SO₄ solution, although with a lower intrinsic activity of Ni in acidic media (**Figure S23a**). Compared with the meagre boost in activity by monophasic Ni(OH)₂ with respect to bare Ni foam, nano-hetero-interfacing decreases the Tafel slope and overpotentials to deliver current densities of -10 and -100 mA cm⁻² significantly (**Figure S23b and c**). Compared with the overpotential difference obtained in alkaline medium, the larger difference in acid signifies the important role of interphasic synergy in boosting the intrinsic activity of Ni (**Figure S24**).

2. Parts of the article are not clearly accessible to the reader and make it difficult for the reader to follow and draw conclusions:

a. lines 296-297 which refer to Figure S45, a detailed legend should be provided (the symbols and the location of the H* and *OH species).

Response. We thank the reviewer for reminding us of the unclear adsorption models in Figure S52. We have provided the back view of the adsorption of *H, *OH, and *HOH species on Ni(OH)₂@NiN₃-C (**Figure R9**) for a clearer illustration of the location of *H, *OH, and *HOH intermediate species and as a clear reference to the text in the paper. A clear legend with symbols of *H, *OH, and *HOH intermediates is provided for Figure S52.

Figure R9. In-plane slab models of (a) *H adsorption on top Ni sites in Ni(OH)₂@NiN₃-C, (b) *H adsorption on bridge Ni sites in Ni(OH)₂@NiN₃-C, (c) *OH adsorption on top Ni sites in Ni(OH)₂@NiN₃-C, and (d) *H-*OH co-adsorption on Ni(OH)₂@NiN₃-C.

Accordingly, Figures S52 is replaced with **Figure R9** in the revised Supplementary Information file, and the following is revised in the new revised manuscript, page 18, lines 352-354:

The adsorption of *H intermediate on the Ni top and bridge sites, *OH intermediate on the Ni top site, as well as the co-adsorption of dissociated *H-*OH on the Ni top and bridge sites, in NiN₃-C phase is modelled over Ni(OH)₂@NiN₃-C (**Figure S52**).

b. The authors conclude that the surface interaction of dissociated H_{ads} and OH_{ads} proceeds preferably at bridge (Ni-C) and top (Ni in NiN₃-C) sites, respectively (line 298-299). Where is the calculation that supports this conclusion? Figure 5b and c present only a visual illustration. Moreover, in the DFT calculations (pages 5-6, SI) the calculations of the hydrogen adsorption energy H* ($\Delta G(H^*)$) is provided without any details for OH* adsorption and the energy barrier for water dissociation H-OH calculations.

Response. We thank the reviewer for raising the point. We have added the related calculations. In order to obtain the preferred adsorption models, we firstly calculated the Gibbs adsorption free energies with two different adsorption sites for H_{ads}. Equations S10 and S11 for hydrogen adsorption energy *H ($\Delta G(*H)$) are described in Methods in the Supplementary Information:

$$\Delta G(*H) = \Delta E + \Delta ZPE - T\Delta S \quad (S10)$$

$$\Delta GE = E\left(\frac{*H}{slab}\right) - E(slab) - 0.5 \times E(H_2) \quad (S11)$$

where ΔE is the binding energy change, ΔZPE is zero-point energy correction, $T\Delta S$ is entropy contribution (at 298 K) to the adsorption free energy $\Delta G(*H)$, $E(H^*/slab)$ and $E(slab)$ are the total energies of *H species adsorption on the slab and the clean slab, respectively. Following the above equations, the calculated $\Delta G(*H)$ values are 1.23 eV for the Ni top site and 0.49 eV for the Ni bridge site. Secondly, we calculated the relaxation of *OH adsorption on the Ni top site in NiN₃-C phase by the following equation:

$$\Delta G(*OH) = \Delta G(*OH/Slab) - \Delta G(Slab) - \Delta G(H_2O) + 0.5 \times \Delta G(H_2) \quad (S12)$$

The obtained $\Delta G(*OH)$ value is 1.744 eV. To further elucidate the favoured adsorption of *H and *OH intermediates on Ni bridge site and Ni top site in NiN₃-C phase, respectively, we performed Bader charge analysis for the adsorption of *H, *OH, and H₂O on the Ni active site in NiN₃-C phase in Ni(OH)₂@NiN₃-C structure (**Figure R10**). Our DFT results show that by considering the *H adsorption alone, the hydrogen atom gains electron (**Figure R10b**). Considering the *OH adsorption alone, hydroxyl is strongly adsorbed on the Ni top site and the oxygen atom gains electron from the Ni site (**Figure R10c**). The adsorption of water also proceeds on Ni top sites where the oxygen atom gains electron from the Ni site (**Figure R10d**). The loss and gain of Bader charges of each adsorbate over Ni active sites in NiN₃-C phase show that the favoured adsorption configuration of *H & *OH intermediates is on Ni bridge sites & top sites, respectively. Furthermore, the plausible pathway for H₂O dissociation on NiN₃-C phase following the co-adsorption of *H-*OH in Figure S54 demonstrate that H_{ads} and OH_{ads} species have similar electron transfer characters and Bader charge variation of *H & *OH suggest electron gain by hydrogen and oxygen atom in hydroxyl. Therefore, the collective observations by adsorption energy and Bader charge variation calculations confirm the favoured adsorption configurations as illustrated in Figure 5b and c.

Figure R10. The loss and gain of Bader charges during the adsorption of (a) *H , (b) *OH , and (d) H_2O on the Ni active site in NiN_3-C phase in $Ni(OH)_2@NiN_3-C$.

To elucidate the related calculations for the preferred adsorption of *H and *OH intermediates as illustrated in Figure 5b and c, the Bader charge analysis in **Figure R10** is added to the revised Supplementary Information file as Figure S51 and the following is added to the revised manuscript, pages 17-18, lines 342-352.

Calculations of the Gibbs adsorption free energies with two different adsorption sites for H_{ads} ($\Delta G(^*H)$) gives 1.23 and 0.49 eV for *H adsorption on Ni top site and Ni bridge site in NiN_3-C phase in $Ni(OH)_2@NiN_3-C$, respectively. Hence, only when a bridge site forged between Ni and C atoms in NiN_3-C phase is considered as the adsorption site, a more optimal hydrogen adsorption energy is attained. In addition, the relaxed *OH adsorption on the Ni top site in NiN_3-C phase via Gibbs adsorption free energy calculation for OH_{ads} gives a value of 1.744 eV. The loss and gain of Bader charges are calculated during the adsorption of *H , *OH , and H_2O on the Ni active site in NiN_3-C phase in $Ni(OH)_2@NiN_3-C$ (**Figure S51**), elucidating that the surface interaction of H_{ads} and OH_{ads} adsorbates proceeds preferably at bridge (Ni-C) and top (Ni in NiN_3-C) sites, respectively (**Figure 5b** and **c**).

In addition, the calculations pertaining to *OH adsorption are added to the section Methods in the revised Supplementary Information file (page 6 and 7). The details on the calculations of the co-adsorption of *H - *OH on the Ni active sites in NiN_3-C phase in $Ni(OH)_2@NiN_3-C$ structure are provided as the response to comment 3 and are also added to the section Methods in the revised Supplementary Information file (page 6 and 7).

3. Lines 302-304: “The adsorption of OH requires an appreciable amount of surface energy (1.74 eV)..... The Gibbs free energy difference slightly increases upon the co-adsorption of *H and *OH species (1.87 eV)” please support these values with the corresponding figure and calculations.

Response. We thank the reviewer for the question. As already explained in the response to comment 2b, the adsorption of *OH on the Ni top site in NiN₃-C phase was calculated by the following equation:

$$\Delta G(*OH) = \Delta G(*OH/Slab) - \Delta G(Slab) - \Delta G(H_2O) + 0.5*\Delta G(H_2) \quad (S12)$$

Similarly, the co-adsorption of *H-*OH on the Ni bridge and top sites in NiN₃-C phase in Ni(OH)₂@NiN₃-C structure was calculated by the definition of Gibbs adsorption free energy and according to following equation.

$$\Delta G(*H-*OH) = \Delta G(*H-*OH/Slab) - \Delta G(Slab) - \Delta G(H_2O) \quad (S13)$$

The outlined 1.74 eV and 1.87 eV for *OH adsorption and *H-*OH co-adsorption, respectively, as well as **Figure R9c** and **d** illustrating the corresponding adsorption models are obtained by the above calculations and elucidate the relaxed adsorption of *OH intermediate on Ni top site as well as the co-adsorption of dissociated *H-*OH species on the Ni bridge and top sites in NiN₃-C phase in Ni(OH)₂@NiN₃-C structure.

Figure R9. In-plains lab models of (c) *OH adsorption on top Ni sites in Ni(OH)₂@NiN₃-C and (d) *H-*OH co-adsorption on Ni(OH)₂@NiN₃-C.

The above calculations are added to the to the section Methods in the revised Supplementary Information file.

4. I agree that the strong H* adsorption on the monophasic Ni(OH)₂ increases the resistance for the subsequent H₂ recombination, however, the H* adsorption free energy (Figure 5) shows that $\Delta G(H^*) = 0.493$ eV for Ni(OH)₂/NiN₃C (bridge), is still not optimal, i.e. far from thermoneutral hydrogen binding energy ($G \sim 0$), please discuss this point.

Response. We thank the reviewer for raising the point. Firstly, we acknowledge that from theoretical point of view, the calculated adsorption energy of hydrogen on Ni bridge sites in NiN₃-C phase in Ni(OH)₂@NiN₃-C structure is still not close to the thermoneutral hydrogen binding energy ($\Delta G \sim 0$). In fact, as described in Figure 5d and in the manuscript, pure NiN₃-C phase has smaller and more optimal adsorption energy (0.26 eV), which is only theoretical. However, that even is not close to the value of 0. Nano-hetero-structuring and the emergence

of interphasic synergy between the two Ni structures accelerate the hydrogen adsorption-desorption-recombination kinetics, and the calculated Gibbs free energy diagram of *H adsorption only predicts that NiN₃-C phase with Ni bridge sites in the nano-heterostructure is more preferred and has more optimal adsorption energy than pure Ni(OH)₂ and other heterostructures including Ni(OH)₂@NiN₃-C with Ni top sites and Ni(OH)₂@NiN₄-C.

In addition to this, the obtained 0.493 eV for Ni bridge sites in NiN₃-C phase in Ni(OH)₂@NiN₃-C heterostructure is based on an ideal model using DFT calculations, where some corrections with only zero-point energy are considered within high accuracy. Herein, the qualitative conclusion is drawn according to the current calculated results. However, some additional effects could be considered for improving theoretical models, such as the important solvent effect (H₂O or ion) and size effect.

According to the raised concern by the reviewer, the above discussions, and to avoid such confusion by the readers, we removed the choice of word “optimal” and revised the lines 361-363 in page 18, as follows:

The Gibbs free energy diagram of *H adsorption predicts NiN₃-C phase with Ni bridge sites in Ni(OH)₂@NiN₃-C heterostructure is more favorable at Volmer step and interfacing with Ni(OH)₂ renders a smaller hydrogen adsorption energy (**Figure 5d**).

5. (a) In Fig S42 and lines 278-283, - the author claimed that the OH_{ads} formation potentials shift to lower potential after HER. If this is the case, I'd expect for changes in the HER LSV curves after 1st, 5th, 10th ... cycles. Please clarify this point.

Response. We thank the reviewer for bringing this point to our attention. We want to state that CV curves in Figure S48 point out to the enhanced OH adsorption on the nano-heterostructure by Ni-N/Ni-C phase (with the pronounced OH_{ads} formation peak for the as-prepared catalyst) as well as the facilitated OH desorption by the synergistic interaction with Ni(OH)₂ phase (with the shifted OH_{ads} formation peak on the reacted surface after HER). It should be specified that all LSV curves presented in HER polarization curves and in supplementary figures are the stable curves of the electrodes. The illustrated CV curves in Figure S48c were obtained on the samples after being tested for HER performance, e.g., after HER polarization curves, where the surface of the catalysts are already reacted under HER potentials. In this context, comparing the electrochemical activation to get polarization curves elucidates both faster stabilization of HER activity with only 5 LSV sweeps and also a sudden shift in the HER onset potential region from the first to second LSV curve for Ni(OH)₂@Ni-N/Ni-C/NF (**Figure R11a**). Whereas it takes more HER activation (13 LSV sweeps) for Ni(OH)₂/NF to attain a stable LSV curve with no significant change (**Figure R11b**). Thus, the interphasic synergy that accelerates surface activation and procures ready-to-serve Ni active sites by facilitating the adsorption-desorption of both *H and *OH intermediates, induce changes in both LSV curves under HER conditions and CV curves at regions excluding HER reaction.

Figure R11. Stabilization of LSV curves for (a) Ni(OH)₂@Ni-N/Ni-C/NF and (b) Ni(OH)₂/NF at 5 mV s⁻¹ in 1 M KOH.

To clarify this point, the changes in LSV curves to get HER polarization curves are demonstrated by adding **Figure R11** as Figure S15 and the following in the revised manuscript, pages 6-7, lines 118-121:

Intriguingly, comparing the electrochemical activation to get HER polarization elucidates both faster stabilization of HER activity and significant shifts between LSV curves at onset and high potential regions for Ni(OH)₂@Ni-N/Ni-C/NF, whereas it takes more activation for Ni(OH)₂/NF to attain stable LSV curves and with no significant change (**Figure S15**).

(b) In addition, in Fig S42c, "after HER" please explain? conditions, potentials ? number of scans?

Response. We thank the reviewer for this notice. "After HER" in Figure S48c refers to CV cycles conducted on the samples that were already tested for HER, i.e., HER polarization. In this case, the surface of Ni(OH)₂@Ni-N/Ni-C/NF and Ni(OH)₂/NF are already activated electrochemically and Ni active sites are subjected to negative potentials. As explained in the response for comment 5a, Ni(OH)₂@Ni-N/Ni-C/NF surface stabilizes with 5 LSV sweeps, showing fast electrochemical activation and ready-to-serve Ni active sites, whereas Ni(OH)₂/NF depicts longer stabilization with 13 LSV sweeps. The LSV test was conducted in three-electrode system with Ni(OH)₂@Ni-N/Ni-C/NF and Ni(OH)₂/NF as working electrode, Hg/HgO as reference and graphite plate as counter electrode, at room temperature in 1 M KOH electrolyte. The potential range applied was between 0 to -0.5 V and scan rate was 5 mV s⁻¹. Afterwards, the reacted samples were collected and used for CV cycles for electrochemical OH adsorption-desorption test, as illustrated in Figure S48c. The potential window was set between -0.05 to 1.3 V, excluding HER reaction and Ni oxidation, and 5 repetitive cycles were recorded at 1 mV s⁻¹. It should be noted that, the electrodes used in Figure S48a and b were as-prepared and the illustrated 500 CV cycles, conducted at 10 mV s⁻¹ represent the surface response of Ni(OH)₂@Ni-N/Ni-C/NF and Ni(OH)₂/NF without Ni oxidation or HER reaction.

In order to avoid ambiguity, we revised the caption for Figure S48 in the revised supplementary Information file as follows:

Figure S48. CV curves of as-prepared (a) Ni(OH)₂@Ni-N/Ni-C/NF and (b) Ni(OH)₂/NF for OH intermediate adsorption at 10 mV s⁻¹ in 1 M KOH. (c) Comparison of CV curves obtained at 1 mV s⁻¹ in 1 M KOH for the electrodes with added Ni ions (Ni(OH)₂@Ni-N/Ni-C/NF_i and

Ni(OH)₂/NF_i) and with Ni from the pristine Ni foam (Ni(OH)₂@Ni-N/Ni-C/NF_p and Ni(OH)₂/NF_p) after HER polarization and obtaining of stable LSV curves.

(c) Fig S42 (a-b) depicts a larger anodic current density at ≈ 1.1 V for Ni(OH)₂@Ni-N/Ni-C/NF compared to Ni(OH)₂, however, a comparison of CV curves after HER (S42c) shows the opposite, with a prominent peak for Ni(OH)₂ at 1.1 V (Fig S42c). Please discuss.

Response. We thank the reviewer for their careful consideration. Considering CV curves conducted on the as-prepared surface of Ni(OH)₂@Ni-N/Ni-C/NF and Ni(OH)₂/NF electrodes in Figure S48a and b, no change is observed for the OH_{ads} formation peak at 1.1 V over Ni(OH)₂/NF, while larger peaks are detected over Ni(OH)₂@Ni-N/Ni-C/NF which weaken to an almost similar intensity observed on Ni(OH)₂/NF. The larger OH_{ads} formation peak at 1.1 V refers to the higher affinity to OH adsorption on Ni-N/Ni-C phase in the nano-heterostructure, while the change in the intensity, which is only observed for Ni(OH)₂@Ni-N/Ni-C/NF, refers to the weakening adsorption of OH charge carriers in alkaline electrolyte as the Ni top sites in Ni-N/Ni-C phase adsorb OH species. Both the larger peak and change in the intensity allude to the oxophilic surface of the nano-heterostructure as a result of nano-hetero-interfacing with Ni-N/Ni-C phase. However, the applied CV potential window excludes HER reaction and *OH intermediates as a result of water dissociation.

During HER reaction conducted prior to the CV test, OH adsorption/desorption is stabilized by both OH charge carriers and *OH intermediates from water dissociation. Hence the effect of charge redistribution between Ni phases in the nano-heterostructure shifts the peak of OH adsorption from alkaline electrolyte during CV cycles in Figure S48c. The intensity of the OH adsorption peak at 0.85 V over Ni(OH)₂@Ni-N/Ni-C/NF is slightly smaller than the intact peak at 1.1 V on Ni(OH)₂/NF. As monophasic Ni(OH)₂ has a strong inherent affinity to OH adsorption but with a slow rate, the OH_{ads} peak remains intact. Whereas the facilitated OH adsorption/desorption due to the accelerated HER kinetics results in the fast weakening of OH adsorption from OH charge carriers in the alkaline electrolyte.

To address the intact and shifted OH_{ads} peaks over Ni(OH)₂/NF and Ni(OH)₂@Ni-N/Ni-C/NF during CV cycles obtained after performing HER reaction over the electrodes, we have added the following in the revised manuscript, pages 16-17, lines 320-331:

This remarks an enhanced oxophilic character and a higher affinity to OH adsorption by nano-hetero-interfacing of Ni-N/Ni-C phase. The larger OH_{ads} peak over Ni(OH)₂@Ni-N/Ni-C/NF eventually weakens to a similar intensity observed on Ni(OH)₂/NF as the adsorption of OH charge carriers in the alkaline electrolyte diminishes gradually. The monophasic Ni(OH)₂ has also a strong inherent affinity to OH adsorption but with a slow rate, and thus, the OH_{ads} peak remains intact.

Intriguingly, OH_{ads} formation shifts to lower potentials over Ni(OH)₂@Ni-N/Ni-C/NF, after reaction under HER potentials (**Figure S15**), where the shoulder peak at 0.85 V becomes prominent (**Figure S48c**). The sluggish OH adsorption over Ni(OH)₂ surface fully retains the OH_{ads} peak after HER, while the accelerated HER kinetics over Ni(OH)₂@Ni-N/Ni-C/NF surface includes OH adsorption/desorption which results in further weakening of OH adsorption from OH charge carriers in the alkaline electrolyte during CV cycles. Thus, a slightly smaller OH_{ads} peak is observed at 0.85 V over Ni(OH)₂@Ni-N/Ni-C/NF than the peak at 1.1 V on Ni(OH)₂/NF.

6. Fig S13(b) should be with ohmic resistance compensation (iR correction). Please check.

Response. We thank the reviewer for this reminder. The Caption of Figure S13 is now corrected specifying the HER polarization curves taken without and with ohmic resistance compensation.

References

- 1 Anantharaj, S. & Noda, S. Amorphous Catalysts and Electrochemical Water Splitting: An Untold Story of Harmony. *Small* **16**, e1905779, doi:10.1002/sml.201905779 (2020).
- 2 Bao, X., Li, Y., Wang, J. & Zhong, Q. Amorphous-crystalline Co–B–P Catalyst for Synergistically Enhanced Hydrogen Evolution Reaction. *ChemCatChem* **12**, 6259-6264, doi:10.1002/cctc.202001343 (2020).
- 3 Lei, C. *et al.* Efficient alkaline hydrogen evolution on atomically dispersed Ni–Nx Species anchored porous carbon with embedded Ni nanoparticles by accelerating water dissociation kinetics. *Energy & Environmental Science* **12**, 149-156, doi:10.1039/c8ee01841c (2019).
- 4 Hong, Y.-R. *et al.* Electrochemically activated cobalt nickel sulfide for an efficient oxygen evolution reaction: partial amorphization and phase control. *Journal of Materials Chemistry A* **7**, 3592-3602, doi:10.1039/c8ta10142f (2019).
- 5 He, S. *et al.* Ni₂P@carbon core-shell nanorod array derived from ZIF-67-Ni: Effect of phosphorization temperature on morphology, structure and hydrogen evolution reaction performance. *Applied Surface Science* **457**, 933-941, doi:10.1016/j.apsusc.2018.07.033 (2018).
- 6 Xu, H. *et al.* Boronization-Induced Ultrathin 2D Nanosheets with Abundant Crystalline–Amorphous Phase Boundary Supported on Nickel Foam toward Efficient Water Splitting. *Advanced Energy Materials* **10**, doi:10.1002/aenm.201902714 (2019).
- 7 Khani, H., Grundish, N. S., Wipf, D. O. & Goodenough, J. B. Graphitic-Shell Encapsulation of Metal Electrocatalysts for Oxygen Evolution, Oxygen Reduction, and Hydrogen Evolution in Alkaline Solution. *Advanced Energy Materials* **10**, doi:10.1002/aenm.201903215 (2019).
- 8 <Operando-EIS.pdf>. doi:10.1021/jacs.0c00257.
- 9 Zhai, P. *et al.* Engineering active sites on hierarchical transition bimetal oxides/sulfides heterostructure array enabling robust overall water splitting. *Nat Commun* **11**, 5462, doi:10.1038/s41467-020-19214-w (2020).
- 10 Zhou, K. L. *et al.* Platinum single-atom catalyst coupled with transition metal/metal oxide heterostructure for accelerating alkaline hydrogen evolution reaction. *Nat Commun* **12**, 3783, doi:10.1038/s41467-021-24079-8 (2021).
- 11 Li, Z. *et al.* Boosting alkaline hydrogen evolution: the dominating role of interior modification in surface electrocatalysis. *Energy & Environmental Science* **13**, 3110-3118, doi:10.1039/d0ee01750g (2020).
- 12 Men, Y., Li, P., Zhou, J., Chen, S. & Luo, W. Trends in Alkaline Hydrogen Evolution Activity on Cobalt Phosphide Electrocatalysts Doped with Transition Metals. *Cell Reports Physical Science* **1**, doi:10.1016/j.xcrp.2020.100136 (2020).
- 13 Zhang, R. *et al.* Selective phosphidation: an effective strategy toward CoP/CeO₂ interface engineering for superior alkaline hydrogen evolution electrocatalysis. *Journal of Materials Chemistry A* **6**, 1985-1990, doi:10.1039/c7ta10237b (2018).
- 14 Li, H. *et al.* Systematic design of superaerophobic nanotube-array electrode comprised of transition-metal sulfides for overall water splitting. *Nat Commun* **9**, 2452, doi:10.1038/s41467-018-04888-0 (2018).
- 15 Liu, B. *et al.* Unconventional Nickel Nitride Enriched with Nitrogen Vacancies as a High-Efficiency Electrocatalyst for Hydrogen Evolution. *Adv Sci (Weinh)* **5**, 1800406, doi:10.1002/advs.201800406 (2018).
- 16 Zhou, P. *et al.* Construction of Nickel-Based Dual Heterointerfaces towards Accelerated Alkaline Hydrogen Evolution via Boosting Multi-Step Elementary Reaction. *Advanced Functional Materials* **31**, doi:10.1002/adfm.202104827 (2021).
- 17 Li, Y. *et al.* Ni-based 3D hierarchical heterostructures achieved by selective electrodeposition as a bifunctional electrocatalyst for overall water splitting. *Electrochimica Acta* **379**, doi:10.1016/j.electacta.2021.138042 (2021).
- 18 Wang, Y. *et al.* Direct Solar Hydrogen Generation at 20% Efficiency Using Low-Cost Materials. *Advanced Energy Materials* **11**, doi:10.1002/aenm.202101053 (2021).

- 19 Zhuang, S. *et al.* The P/NiFe doped NiMoO₄ micro-pillars arrays for highly active and durable hydrogen/oxygen evolution reaction towards overall water splitting. *International Journal of Hydrogen Energy* **44**, 24546-24558, doi:10.1016/j.ijhydene.2019.07.138 (2019).
- 20 Xin, Y., Kan, X., Gan, L. Y. & Zhang, Z. Heterogeneous Bimetallic Phosphide/Sulfide Nanocomposite for Efficient Solar-Energy-Driven Overall Water Splitting. *Acs Nano* **11**, 10303-10312, doi:10.1021/acsnano.7b05020 (2017).
- 21 Wang, P. *et al.* MnO_x-Decorated Nickel-Iron Phosphides Nanosheets: Interface Modifications for Robust Overall Water Splitting at Ultra-High Current Densities. *Small* **18**, e2105803, doi:10.1002/sml.202105803 (2022).
- 22 Liu, T. *et al.* Interfacial Electron Transfer of Ni₂P-NiP₂ Polymorphs Inducing Enhanced Electrochemical Properties. *Adv Mater* **30**, e1803590, doi:10.1002/adma.201803590 (2018).

Reviewer comments, third round review -

Reviewer #1 (Remarks to the Author):

The authors have carefully addressed all concerns in the revised manuscript and acceptance is recommended.

Reviewer #2 (Remarks to the Author):

The authors have addressed suggestions and requests for clarification in full. Subsequently, the reviewer finds the article after revision at the appropriate level and suitable for publication in Nature Communications.

Reviewer #3 (Remarks to the Author):

The authors have made the appropriate changes and provided clear explanations supported by further experimental work in the revised manuscript. This work is therefore recommended for publication in Nature Communications.